# Rapid construction of a whole-genome transposon insertion collection for *Shewanella oneidensis* by Knockout Sudoku

Michael Baym[1,*], Lev Shaket[2], Isao A. Anzai[2], Oluwakemi Adesina[2] & Buz Barstow[2,*]

Whole-genome knockout collections are invaluable for connecting gene sequence to function, yet traditionally, their construction has required an extraordinary technical effort. Here we report a method for the construction and purification of a curated whole-genome collection of single-gene transposon disruption mutants termed Knockout Sudoku. Using simple combinatorial pooling, a highly oversampled collection of mutants is condensed into a next-generation sequencing library in a single day, a 30- to 100-fold improvement over prior methods. The identities of the mutants in the collection are then solved by a probabilistic algorithm that uses internal self-consistency within the sequencing data set, followed by rapid algorithmically guided condensation to a minimal representative set of mutants, validation, and curation. Starting from a progenitor collection of 39,918 mutants, we compile a quality-controlled knockout collection of the electroactive microbe *Shewanella oneidensis* MR-1 containing representatives for 3,667 genes that is functionally validated by high-throughput kinetic measurements of quinone reduction.

[1] Department of Systems Biology, Harvard Medical School, Boston, Massachusetts 02115, USA. [2] Department of Chemistry, Princeton University, Princeton, New Jersey 08544, USA. * These authors contributed equally to this work. Correspondence and requests for materials should be addressed to B.B. (email: buz@princeton.edu).

Shewanella oneidensis MR-1 is one of the archetypal members of the electroactive microbes[1]. Members of this class are capable of transferring electrons to, from, or between metabolism and a remarkable variety of external substrates, ranging from metal ions[2] to solid surfaces including electrodes[1]. The *Shewanellae* serve as model organisms for understanding the role of microbes in the biogeochemical cycling of metals[3] and carbon[4] as well as the geochemical evolution of the Earth[5,6]. *S. oneidensis* possesses an extensive complement of *c*-type cytochrome electron transfer proteins[7], including the *mtr* extracellular electron transfer (EET) operon that encodes a set of proteins responsible for most of the electron flux from metabolism to external electron acceptors[4]. This cytochrome network provides *S. oneidensis* with extraordinary respiratory flexibility[8] and gives it potential applications in environmental remediation[3,5], nuclear stewardship[9] and sustainable energy[10,11], while its role in the cryptic cycling of sulfur may have consequences for the fate of $CO_2$ sequestered in deep aquifers[12]. However, despite intense research, the functions of over half of its genes remain unknown[13] and fundamental discoveries about its nature continue to be made[12,14]. Although the centrality of the MtrABC complex in EET is well established[15], debate continues about the full complement of proteins used in this process, particularly in electron transfer from the inner membrane, across the periplasm, to the outer membrane[16–19]. There remains an ongoing need for the development of new, highly accessible tools for the genetic manipulation and characterization of *S. oneidensis*[1,8,20] and the menagerie of esoteric microorganisms found in nature that offer unique capabilities to biological engineering and medicine.

Aside from the genome sequence, a single-gene knockout collection is one of the most valuable genetic tools for any organism. Transposon mutagenesis, in which a defined DNA sequence is randomly integrated into a single genomic site, allows the straightforward creation of extremely large gene disruption mutant libraries for a wide variety of microorganisms[21,22].

Recently developed techniques that use next-generation sequencing (NGS) of pooled transposon mutant libraries[23–25] have made dramatic advances in the characterization of genes that can be selected for fitness contributions[26] to species including *S. oneidensis*[13,27,28]. Nevertheless, clonally isolated collections of mutants remain critically important for the characterization of phenotypes such as virulence factors[29,30], secondary and cryptic metabolite production[31] and behaviours synonymous with *Shewanellae*, such as biofilm formation[32,33] and EET[34].

Conventionally, a small collection (5,000–8,000 members) of transposon mutants is constructed solely for a specific genetic screen, hits are identified and isolated while the rest of the collection is often discarded. The random nature of transposon mutagenesis means that unless these non-catalogued collections are extremely large (many times the organism's gene count), then representative mutants for many non-essential genes will be missing[35]. Additionally, these collections do not facilitate testing of specific gene function predictions[36].

The utility of comprehensive, non-redundant, catalogued (curated) collections is hard to overstate: in the 15 years since the release of the Yeast Knockout Collection (YKO) of *S. cerevisiae* gene deletion mutants, it was used in over 1,000 genome-wide experimental screens[37]. While curated collections made by targeted gene deletion remain the gold standard[37], their construction requires an extraordinary technical and financial undertaking, and as a result, only a handful exist[23,35,38–41] (Supplementary Note 1). Cataloging extremely large transposon mutant collections by Sanger sequencing and condensation[30,42,43] significantly reduces costs, but is still prohibitive (Supplementary Note 1).

NGS has led to an explosion of genomic data in the past decade. This has inspired the development of techniques that dramatically reduce the cost of cataloging transposon mutant collections by combinatorial pooling, followed by a single-NGS experiment, and analysis of the resulting data set[24,26]. The problem solved in this analysis, reconstructing the location of genomic coordinates on a grid of wells, resembles the number placement puzzle Sudoku[44]. This exciting development has facilitated the construction of a growing number of curated collections for pathogens and pathogen surrogates[36,45,46]. However, the costs, complexity and low speeds of the robotic hardware needed to combine samples before sequencing make these methods inaccessible to many investigators (Supplementary Note 1). This limitation has inspired schemes that do not rely on robotics for sample preparation[47]. However, the data sets produced by these manual pooling methods pose significant challenges for reconstruction due to the presence of non-unique sequences that map back to both their originating locations and to many other artifactual ones[44].

Here, we describe a low-cost, easily implemented, rapid and generalizable method for automatically cataloging extremely large transposon mutant collections and then guiding the construction and validation of a condensed, quality-controlled (QC) collection. We have developed an algorithm that uses self-consistency within a sequencing data set to disambiguate the location of non-unique transposon mutants. This allows the use of an extremely rapid, simple manual combinatorial pooling method that is able to pool a 39,918 member progenitor *S. oneidensis* transposon mutant collection, suitably large to contain a mutant for a large fraction of non-essential genes in the *S. oneidensis* genome, in a single day (Fig. 1 and Supplementary Fig. 1). By contrast, a robotic method would require almost 100 days of pooling at a cost of approximately $30,000 in user fees (Supplementary Note 1). The progenitor collection catalogue is used to algorithmically guide the re-array and colony purification of a non-redundant set of mutants. This collection is then independently validated through a second round of combinatorial pooling, sequencing, and orthogonal sequence analysis followed by a curation step for QC. In total, the QC collection comprises isolated mutants for 3,667 genes. Furthermore, we functionally validate this collection through a screen for reduction of anthraquinone-2,6-disulfonate (AQDS), an analogue for the redox active moieties found in humic substances that has been shown to enhance EET to minerals by dissimilatory metal reducing bacteria[48]. Using high-throughput photography, time courses of AQDS reduction are captured nearly simultaneously for every mutant in the collection, recapitulating previous discoveries in a single experiment as well as verifying the functions of associated genes. With the wealth of sequencing data now available for both the *Shewanellae*[49] and other much more esoteric organisms, Knockout Sudoku provides an exciting avenue for comparative genomics and systems biology.

## Results

**Model of transposon insertion.** Transposon insertion is a random, independent, one time per microbe event that can be assumed to have approximately uniform probability across the genome[50]. The number of mutants in the progenitor collection needed to find a representative knockout for every non-essential gene was estimated by two methods: a commonly used analytical Poisson-statistics-based formula[51]; and a Monte Carlo numerical simulation, both of which relate the number of genes with one or more transposon insertion mutants (the represented gene count) in the collection to its size (Methods).

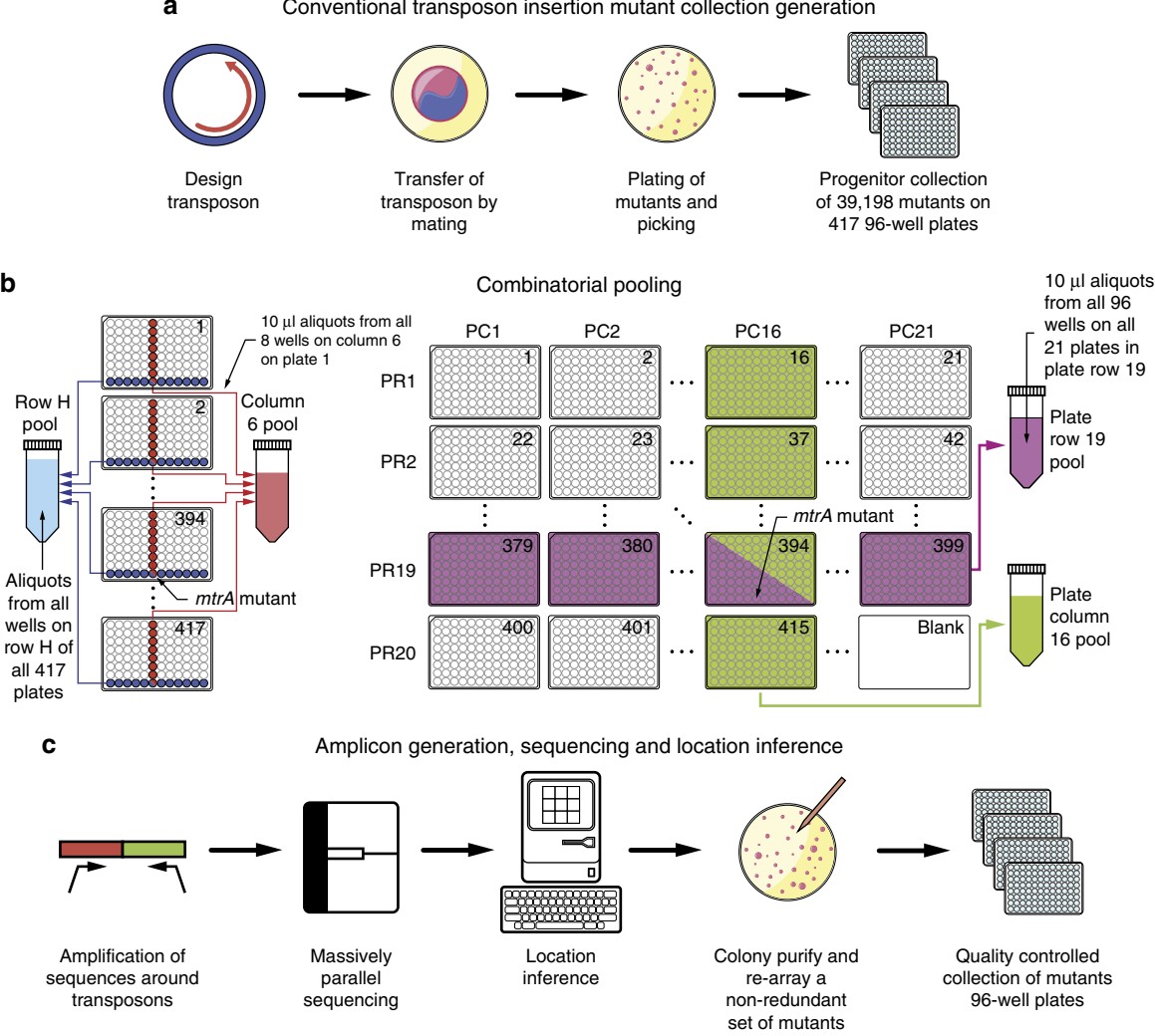

**Figure 1 | Workflow for creation of the quality-controlled *S. oneidensis* Sudoku collection by Knockout Sudoku. (a)** A massively oversampled mutant collection is created by conventional transposon mutagenesis. (**b**) Combinatorial pooling is used to prepare mutant libraries for barcoding. (**c**) Mutants are located by construction of a next-generation sequencing library by amplicon generation and barcoding, next-generation sequencing, mutant location inference, mutant selection and construction of a quality-controlled mutant collection.

The genome can be grouped into three categories: non-essential genes (Fig. 2a; regions NE1 and NE2); essential genes (Fig. 2a; E1 and E2); and intergenic regions (Fig. 2a; I1 and I2). The total number of non-essential genes for growth of *S. oneidensis* on LB (NE1 + NE2) is estimated to be between 3,256 and 4,184 (ref. 13) (Methods). Most non-essential genes can accept transposon insertions (NE1), as can many intergenic regions without a fatal loss of organism fitness (I1). There will likely be some non-essential genes into which transposon insertion is a low to zero probability event due to the scarcity or absence of the insertion site (NE2) (an AT or TA dinucleotide for the mariner system used here[52]). Conversely, there will be a small number of essential genes that can accept transposon insertions, for instance at the end of the coding region[28] (E1), whereas insertion at any point in most will be fatal (E2). Finally, there will likely be intergenic regions in which the transposon insertion site is scarce or absent or in which insertion will be fatal (I2).

The Poisson-model only considers non-essential genes and assumes that they are all equally likely to accept transposons (Methods). In this model, the represented gene count asymptotes to the total non-essential gene count (NE1 + NE2) most rapidly:

26,335 colonies for 3,256 genes and 34,890 colonies for 4,184 genes (Fig. 2b, red and blue dashed lines).

By contrast, a Monte Carlo simulation allows transposon integration at any acceptable site in non-essential genes or intergenic regions (Methods). Although insertion into an intergenic region yields a collection member ($x$ axis), it does not increase the represented gene count ($y$ axis), which thus rises more slowly than in the Poisson-model (Fig. 2c). In addition, the represented gene count reaches a quasi-plateau slightly below the non-essential gene count, reflecting the existence of non-essential genes into which transposon insertion is improbable or impossible (NE1). For the cases of both 3,256 and 4,184 non-essential genes, the represented gene count does not begin to plateau until ~40,000 mutants (Fig. 2c).

**Progenitor collection construction.** The first steps of Knockout Sudoku follow those of conventional transposon mutant collection generation using a modified pMiniHimar system[21] (Fig. 1a; Methods; Supplementary Data 1). Based upon consideration of the Monte Carlo model of transposon insertion, and our estimate

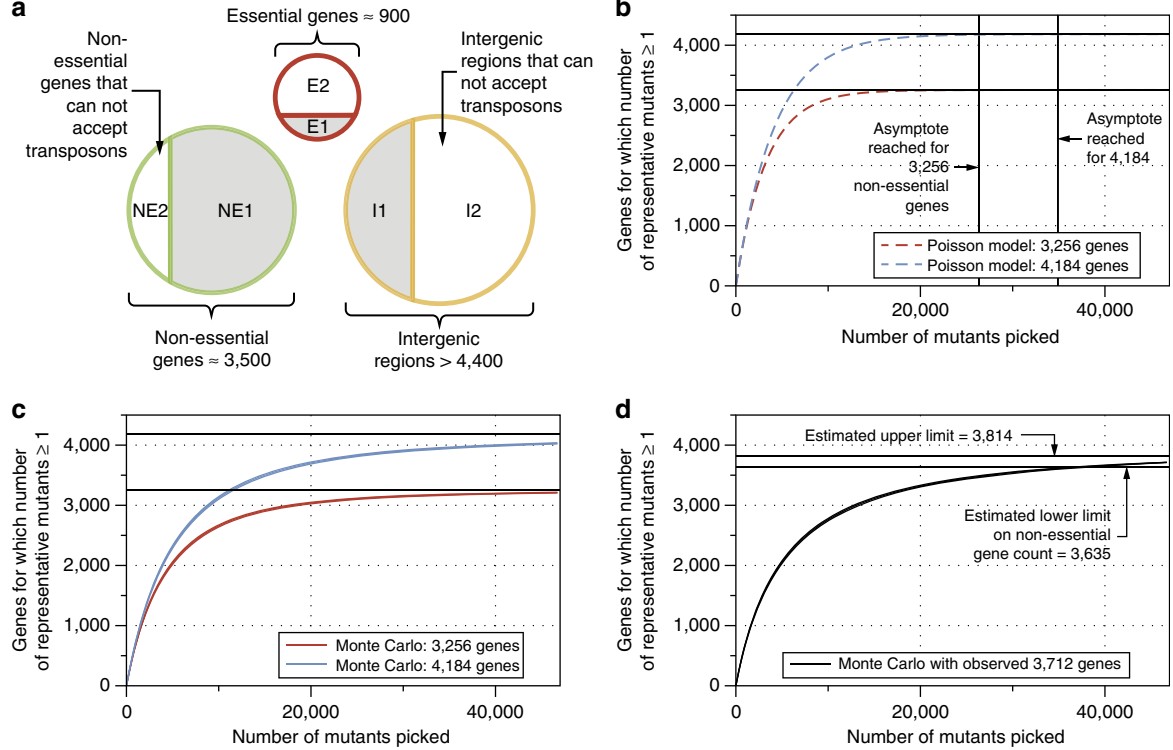

**Figure 2 | Model of transposon insertion.** (**a**) Genome feature type breakdown for *S. oneidensis*. Regions of the genome (essential genes, non-essential genes and intergenic regions) into which transposons can be inserted (E1, NE1 and I1) are coloured grey. (**b**) Estimate number of genes for which the number of representative mutants ≥1 (represented gene count) as a function of mutant collection size by a Poisson model for genomes with 3,256 (red lines) and 4,184 (blue lines) non-essential genes. (**c**) Estimate of represented gene count by a Monte Carlo model for genomes with 3,256 (red curves) and 4,184 non-essential genes (blue curves). The centre of the solid curves is the mean value of the unique gene disruption count from 1,000 simulations while the upper and lower curves represent two standard deviations around this mean. (**d**) Estimate of represented gene count using a random drawing without replacement from the observed set of transposon insertion mutants. The centre of the solid curves is the mean value of the unique gene disruption count from 1,000 simulations, while the upper and lower curves represent two s.d.s around this mean.

of the number of mutants that we could pick and pool without risking sample perishment, we picked a progenitor collection of 39,198 *S. oneidensis* transposon insertion mutants over 2 days (Methods).

**Combinatorial pooling**. The goal of combinatorial pooling is to combine many samples into a single-NGS experiment and then reconstruct the physical locations of each sequence. We developed a four-dimensional combinatorial pooling scheme that can be easily performed with multi-channel pipettors while minimizing sample preparation time and cost (Fig. 1b). All 417 plates in the progenitor collection were assigned to a position within a virtual 20 × 21 grid (Fig. 1b). The assignment of plate-row (PR) and plate-column (PC) coordinates to each plate (rather than one) allows the cost of sequencing library construction to grow only with the square root of the number of plates. Aliquots of culture from each well in the collection were dispatched to four pools that uniquely corresponded to the coordinates of the well. For example, a mutant with a disruption in the *mtrA* cytochrome that was located in well H6 on plate 394 was dispatched to the Row H, Column 6, PR 19, and PC 16 pools (Fig. 1b). In contrast, robotically actuated pooling schemes dispatch samples to a set of pools that do not directly correspond to their physical location[24,44]. In total, we filled 61 address pools (20 PR × 21 PC pools, 8 row × 12 column pools). The entire progenitor collection was pooled and cryopreserved in a single day using a 96-channel pipettor and a team of five people (Methods). The address pools were used to generate 61 barcoded amplicon libraries that

encoded the genomic locations of the transposons in each pool and were sequenced in a single-NGS experiment (Methods).

**Transposon mutant location**. To locate mutants within the progenitor collection, the sequencing data set was parsed to a set of transposon insertion locations and pool address coordinates. Figure 3a shows an annotated read from an amplicon produced by the *mtrA* disruption mutant highlighted in Fig. 1. A flow chart of the sequence analysis workflow can be found in Supplementary Fig. 1. Overall, of the 146,128,068 initial reads generated by Illumina, 120,085,483 aligned to genomic locations and contained valid barcode and transposon sequences (Methods; Supplementary Fig. 1, algorithm 2, KOSUDOKU-SEQANALYZE). For each unique genomic location, we counted the number of reads with a given barcode to construct a pool presence table in order to deduce mutant locations (Fig. 3b). Out of the 83,380 unique entries in the pool presence table, roughly half corresponded to low-frequency amplification events that were unlikely to be due to actual transposon insertions in the genome: for example, 19,723 entries were associated with only a single read. These events were systematically filtered by requiring a threshold of five read counts per coordinate, leading to a stable solution of the pool presence table with a physically reasonable number of entries (39,588) that had one or more coordinates that could be used for mapping (Supplementary Fig. 3). The read counts associated with all coordinates in each axis (namely the row, column, PR and PC pools) followed an approximately log-normal distribution (Supplementary Fig. 4). A summary of the sequence analysis steps is shown in Fig. 3i.

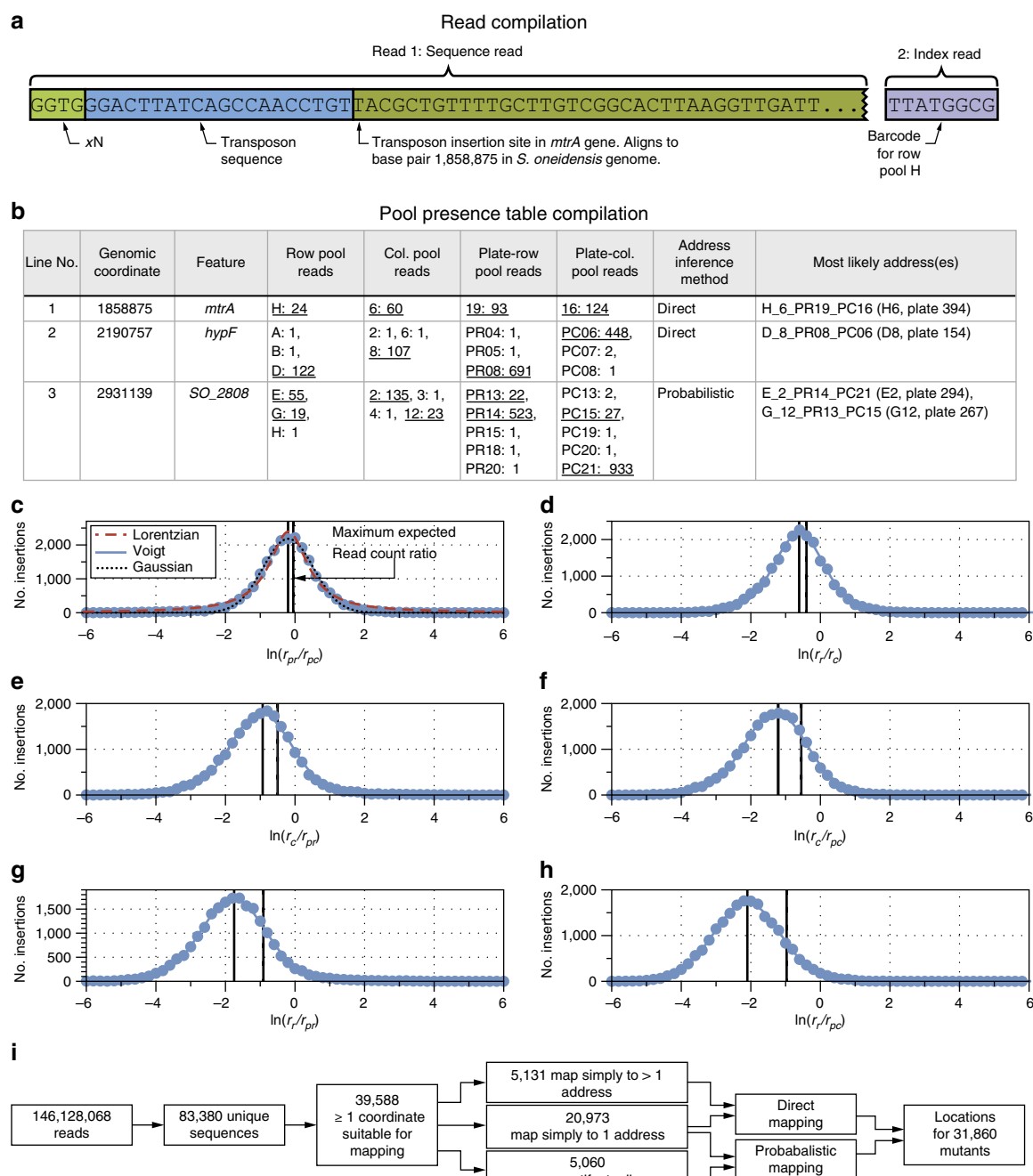

**Figure 3 | Knockout Sudoku sequence data analysis. (a)** Reduction of read data to transposon insertion coordinates and pool barcode identities. (**b**) Compilation of pool presence table that summarizes pool barcodes of reads aligning to each genomic coordinate. Validations of predictions in lines 2 and 3 can be found in Supplementary Data 3. (**c–h**) Distribution of read count ratios for pool presence table lines that unambiguously map to single addresses (line 1). (**c**) Plate-row (pr) to plate-column (pc) and comparison of log-Voigt, Gaussian and Lorentzian fits for the read count ratio distribution; (**d**): row (r) to column (c); (**e**) column to plate-row; (**f**) column to plate-column; (**g**) row to plate-row; (**h**) row to plate-column. The maximum read count ratios that would be expected for uncorrelated underlying read count distributions are marked as vertical dashed lines in **c–h** (Supplementary Data 2). (**i**) Condensation of sequencing reads to mutant addresses.

Line 1 of the pool presence table samples in Fig. 3b corresponds to the *mtrA* disruption mutant highlighted in Fig. 1b and simply maps to well H6 on plate 394. Just over half (20,973) of the sequences in the pool presence table map in a similarly straightforward manner. In addition, another 5,131 sequences are non-unique but map to multiple addresses without any artifacts. This type of entry can arise if a sequence is found in multiple places that share all but one coordinate, if for example the same colony was picked into two wells in the same row of a

single plate. A further 8,424 entries did not have a complete set of coordinates.

The remaining 5,060 entries had multiple coordinates that combined to produce both the original and artifactual addresses. For example, the coordinates shown in Fig. 3b (line 3) most likely originate from only 2 addresses but form 16 possible combinations. Unlike in a robotically pooled scheme, to allow the use of multi-channel pipetting and save time, our simple scheme builds no additional data into the pool presence table that allows

coordinates to be immediately connected from axis to axis such as a checksum[24,44]. Examining simply the presence or absence of a sequence within a pool cannot disambiguate the location of repeat insertions[24,44]. However, the additional richness of the sequencing data set, namely the read counts, allows for the connection of sets of coordinates and deduction of the location of non-unique mutants (Methods).

*A priori*, we expect that the frequency of a given DNA amplification event should not vary significantly from pool to pool and that the ratio of the reads from multiple pools should be fixed around a mean value. Thus, we would expect that the entry in Fig. 3b (line 3) is made up of two sets of coordinates: one with high read counts and a second with low read counts. For any given mutant from any location, we expect that the ratios of read counts will on average reflect the ratios of template of that mutant loaded into the amplicon generation reactions. For example, the amount of template loaded into a column pool reaction should be on average 12/8th of that in a row pool reaction (Supplementary Note 2). Thus, it is reasonable to expect that on average the number of reads observed for any given mutant in a column pool will be at least 12/8th of that in the corresponding row pool (Supplementary Note 2; Supplementary Fig. 4).

The effect of amplification frequency conservation across the sequencing data set can be understood by modelling any one of the six ratios of read counts for the four coordinates (R, C, PR, PC) by Monte Carlo simulation. We found that the expected distribution of read count ratios was best fit by a log-normal function whose centre moved to lower values as both the ratio of mean values of the underlying read count distributions was decreased and the correlation between them was increased (Supplementary Fig. 5 and Supplementary Note 3).

To test this expectation, we calculated histograms of the six read count ratios for all of the pool presence table entries that mapped to single addresses (Fig. 3c–h). For example, the read count ratios for Fig. 3b, line 1 are $r_{pr}/r_{pc} = 93/124$ (plate row to plate column); $r_r/r_c = 24/60$ (row to column); $r_c/r_{pr} = 60/93$; $r_c/r_{pc} = 60/124$; $r_r/r_{pr} = 24/93$ and $r_r/r_{pc} = 24/124$. As anticipated, the centres of these distributions move to lower values as the ratio of individual mutant template used in the amplicon library generation reaction decreases (Supplementary Notes 2 and 3). These centres are lower than would be expected for uncorrelated underlying distributions (Supplementary Fig. 5 and Supplementary Note 3) and were approximately fit by log-normal functions (Fig. 3c–h, black dashed curves). However, we found that the distributions were most accurately fit by a log-Voigt function (Fig. 3c–h, blue curves) that is typically used to describe distributions that deviate slightly from both log-Lorentzian and log-normal. We speculate that this is due to deviations of the underlying read count distributions from log-normal at low read counts (Supplementary Fig. 4).

The log-Voigt functions were converted to a set of probability density functions. This permitted us to rate the probability that a given combination of coordinates contains the mutant by simply multiplying together the probabilities of the read count ratios for a given combination of coordinates in one of the proposed addresses for a non-unique sequence.

In total, we mapped 31,860 mutants to 21,907 singly occupied and 9,922 multiply occupied wells in the progenitor collection. The remaining 7,369 wells are a likely home for the mutants with insufficient address data. Histograms of the density of mappable transposons in the progenitor collection across the *S. oneidensis* chromosome and megaplasmid are shown in Fig. 4 along with a plot showing the location of individual transposons in the vicinity of the *mtr* EET operons. As seen in earlier studies, transposon insertion occurs preferentially near the origin of replication of the

chromosome[50], whereas the megaplasmid is sufficiently short that this effect is not readily apparent.

Overall, we found disruption mutants for 3,474 loci in singly occupied wells while mutants for 238 could only be found in multiply occupied wells (a total of 3,712 loci). A complete catalogue of the predicted progenitor collection contents can be found in Supplementary Data 2. The predicted sequence identities of 174 unique progenitor collection members were verified: 163 were accurate to within 1 bp, while the maximum error of the remainder was 5 bp (Methods; Supplementary Data 3).

**Generation and validation of quality-controlled collection**. The progenitor collection catalogue was used to assemble a non-redundant QC collection of 3,630 mutants that disrupt a total of 3,667 genes out of the 4,587 genes in the *S. oneidensis* genome. Mutants for 45 genes had not been isolated from the progenitor collection at the time of writing. Oversampling not only ensures sufficient coverage of the genome but also allows us to choose the mutant that best disrupts a given gene. In order to minimize overall collection construction time and effort, we developed a mutant selection algorithm that balances likelihood of disruption with ease of isolation from well co-occupants.

The mutants chosen to represent each of the genes in the *mtr* operons and surrounding genome are highlighted in red in Fig. 4. In most cases, the collection condensation algorithm selects the closest transposon mutant to the translation start of the gene. However, in a number of cases, such as *mtrD*, the algorithm chooses a mutant further into the coding region of the gene as earlier mutants share wells with a large number of co-occupants (Methods).

We found no mutants for 875 genes in the *S. oneidensis* genome. In many cases this reflects their essentiality; for instance, the cysteine synthesis gene *cysS*, and the ferrous iron transporter protein encoding genes *feoA* and *feoB* (ref. 27) all belong to the E2 (Fig. 2a) group of essential genes which contain no acceptable transposon insertion sites.

There are a number of genes that are essential but have representative mutants (Fig. 2a, E1). For instance, a disruption at the C-terminal end of the folate synthesis gene *folD* is shown in Fig. 4. The progenitor and condensed collections also include a mutant for the anaerobic ribonucleotide reductase β-subunit (*nrdB*). Previous studies have shown that the C-terminal residues of *nrdB* are essential for interactions with its neighbouring α-subunit[53]; however, the transposon in our collections inserted between the last two base pairs (within the stop codon), missing these essential residues.

We generated a condensed collection by re-array of 2,699 representative mutants from singly occupied wells, and colony purified representatives for another 999 loci from multiply occupied wells. For the samples from multiply occupied wells, we picked between 2 and 10 colonies into a series of 96-well plates based on the predicted content of the well. In total, this collection comprised 5,653 samples on 84 plates. The collection was arranged into a $9 \times 10$ plate grid, re-pooled and resequenced (Methods; Supplementary Fig. 6).

We used an orthogonal sequence validation algorithm to test if the predicted content of each well in the condensed collection was present (Methods; Supplementary Fig. 6, algorithm 7, KOSUDOKU-ISITINTHERE). Out of the 5,653 mutant identity predictions, 5,202 were correct, a 92% accuracy rate that indicates success in sequence location, initial re-array, colony purification and picking. It is important to note that a finite error rate in re-arraying should be expected: the first generation Keio *Escherichia coli* whole-genome knockout collection[39] has a 4% error rate[54]. However, a key advantage of the low cost and convenience of

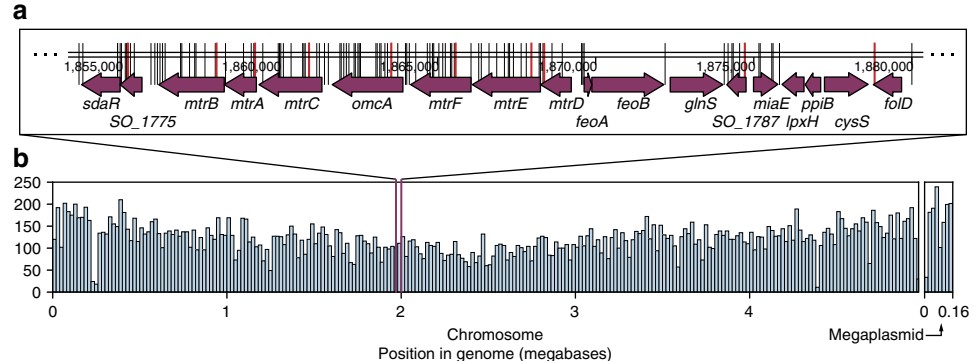

**Figure 4 | Location of transposon mutants in progenitor collection.** (**a**) Map of locations of transposons in the progenitor collection (black lines) and those chosen for the quality-controlled collection (red lines) in the vicinity of the *mtrCAB* and *mtrDEF* extracellular electron transport operons. It is important to note that the chosen mutant was not always the one closest to the translation start, as in a number of cases our selection algorithm (Methods) estimated that the benefits to gene disruption were outweighed by the difficulties of isolation. (**b**) Histogram showing the density of transposons in the progenitor *S. oneidensis* Sudoku collection (bin width is 20,000 base pairs).

Knockout Sudoku is the ability to immediately diagnose this error. In total, we verified the identity of 3,245 mutants, disrupting 3,282 loci. In addition, we later added 385 mutants by colony purifying samples from the progenitor collection over the course of ~2 days, bringing the total of disrupted loci to 3,667. With this extension, the QC collection contains a representative for almost 99% of the loci disrupted in the progenitor collection. A complete catalogue of the QC collection can be found in Supplementary Data 4.

To gauge the completeness of the progenitor collection, we constructed a redundant list of locatable sequences from the progenitor collection (for example, if a mutant appears once in the progenitor collection its transposon location appears once in this list; if it appears 10 times, the location appears 10 times). We used this list to perform a Monte Carlo simulation (similar to the collection construction simulation), in which we randomly picked from the locatable sequences and calculated the represented gene count (Methods) (Fig. 2d, black lines). The represented gene count rises at a similar rate to those in earlier simulations, perhaps raised slightly by the lethality of some intergenic insertions. However, the plateau point is determined not only by the number of non-essential genes with transposon insertion sites (NE2) but also by the number of essential genes into which transposition is possible (E2), a larger set than those considered earlier.

We estimate the number of non-essential genes for LB growth to be 3,712 (between 3,635 and 3,814), with the upper bound coming from our estimate of the non-essential genes with low to zero transposon insertion probability (NE1), and the lower bound coming from our estimate of the essential genes with transposon insertions in the progenitor collection (E2, for example, *folD*, *nrdB*) (Methods). This estimate is larger than the 3,355 non-essential genes reported by Deutschbauer *et al.*[13] due to a larger progenitor collection, and understandably larger than the 2,216 reported by Yang *et al.*[28] and the 3,030 gene fitness values provided by Brutinel *et al.*[27] due to the more permissive growth conditions of LB relative to *Shewanella* Basal Media.

**Functional validation of the quality-controlled collection**. We screened the QC collection for reduction of anthraquinone-2,6-disulfonate (AQDS), an analogue for the redox active moieties found in humic substances that can act as electron acceptors during anaerobic respiration[55]. AQDS changes colour from clear to deep orange when reduced by two electrons, allowing visual identification of mutants that are unable to reduce it. For this

screen, a small aliquot of saturated culture was transferred to an assay plate containing *Shewanella* Basal Media and AQDS. Although lactate is normally added as an electron donor in this type of experiment[55], we omitted it here because our earlier experiments showed that *S. oneidensis* is still capable of reducing AQDS in its absence. A time series of images was captured for every plate in the collection over 40–60 h under an anaerobic atmosphere (97% $N_2$, 3% $H_2$) and was automatically reduced to a time course for each mutant in the collection (Figs 5 and 6 and Supplementary Figs 7 and 8; Methods). The linear portion of each time course (Supplementary Fig. 7) was fitted to calculate a reduction rate (Fig. 6d; Supplementary Data 6).

Our screening results include previously identified genes as well as additional associated genes. Mutants with disruptions in the *menC* menaquinone synthesis gene[55]; the *tolC*-like efflux pump *SO_2917* that moderates AQDS toxicity[56]; and the *ccmC* cytochrome maturation gene[21] all showed large deficiencies in their ability to reduce AQDS (Fig. 5c–e).

In addition, a dramatic loss of AQDS reduction ability was registered for disruption mutants of the menaquinone synthesis pathway members *menA*, *B*, *D*, *E* and *F* and the *c*-type cytochrome maturation operon members *ccmA*, *B*, *D*, *E* and *G* (Fig. 5d,e).

In agreement with earlier measurements, disruption of the MtrB outer-membrane porin, which supports outer-membrane metal reductase MtrC, slows AQDS reduction[56] to 42% of the quasi wild-type rate. Similarly, disruptions of the type II secretion system genes *gspC*, *gspM* and *gspN*, which are responsible for transporting MtrB to the outer membrane, considerably slow AQDS reduction[57] (Supplementary Fig. 7B).

While the disruptions to the *ccm* operon indicate a crucial role for cytochromes, such as those encoded by the *mtr* genes, in electron transfer to AQDS (Fig. 5e), no disruption in any of the genes in the *tor* (TMAO reductase); *sir* (sulfite reductase); and *dms* (DMSO reductase) operons produced any noticeable diminution of AQDS reduction (Supplementary Fig. 8 and Supplementary Data 6).

**Additional functional observations**. We also noted a number of new observations that are worthy of discussion. As in Fe(III) reduction, disruptions of the genes coding for MtrA and MtrC[58] and for the inner membrane *c*-type cytochrome CymA[15,59] which is a key distribution point for electrons to the MtrABC complex, slow AQDS reduction to 80%, 83% and 70% of the quasi wild-type rate respectively (Fig. 6d). Consistent

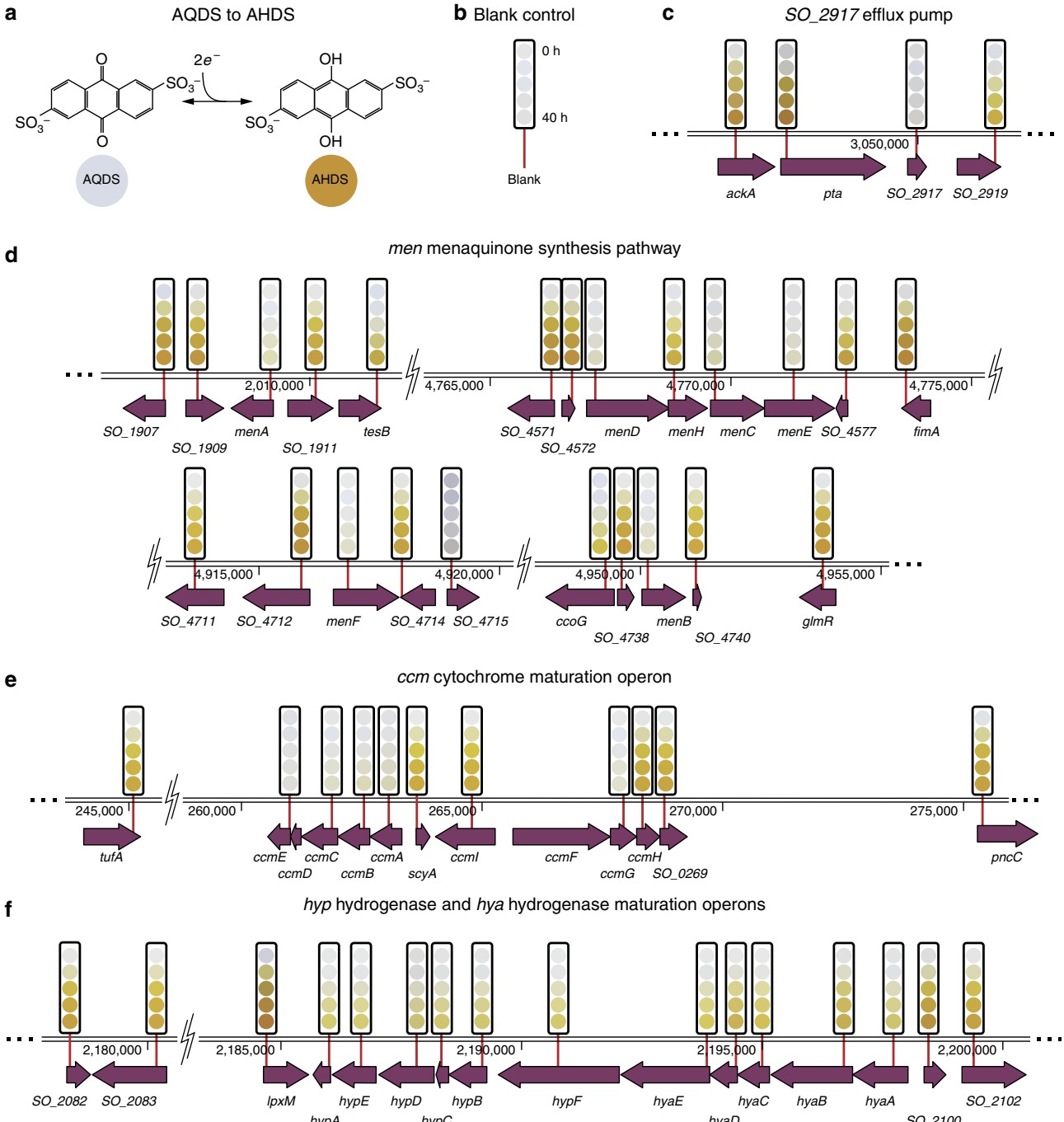

**Figure 5 | Gene disruptions that change AQDS reduction rate. (a)** AQDS (anthraquinone-2,6-disulfonate) redox reaction. AQDS changes colour from clear to orange when reduced. **(b)** Blank control. **(c–f)** Selected gene disruptions that produce changes to AQDS reduction rate that are observable to the unaided eye. The state of the AQDS dye at 10 h intervals is indicated by a series of coloured circles above the location of the transposon chosen to disrupt each gene (Methods).

with reports that disruption of the *mtrDEF* operon, paralogous to the *mtrCAB* operon, has no role in Fe(III) reduction[18], disruptions here had no discernible impact on AQDS reduction (Fig. 6a).

The necessity of the TolC efflux pump for mutant survival suggests that a large fraction of interactions with AQDS occur in the periplasm and perhaps in the cytoplasm. However, the weak but detectable dependence of AQDS reduction rate on the MtrA, MtrC and CymA cytochromes could indicate that interactions also occur outside of the cell. While these results need to be

validated by targeted deletion to produce a clear genotype–phenotype connection and eliminate any possible polarity effects on downstream genes, their significance could suggest some role for MtrA, MtrC and CymA in AQDS reduction.

Disruption of any of the genes encoding the Hyp NiFe uptake hydrogenase, *hypA, B, C, D* and *E*, and its maturation factors *hyaA, B, C, D* and *E* lead to significant loss of AQDS reduction rate and completeness (Fig. 5f). These results prompt us to speculate that in the absence of lactate, oxidation of $H_2$ can be a significant electron source for EET.

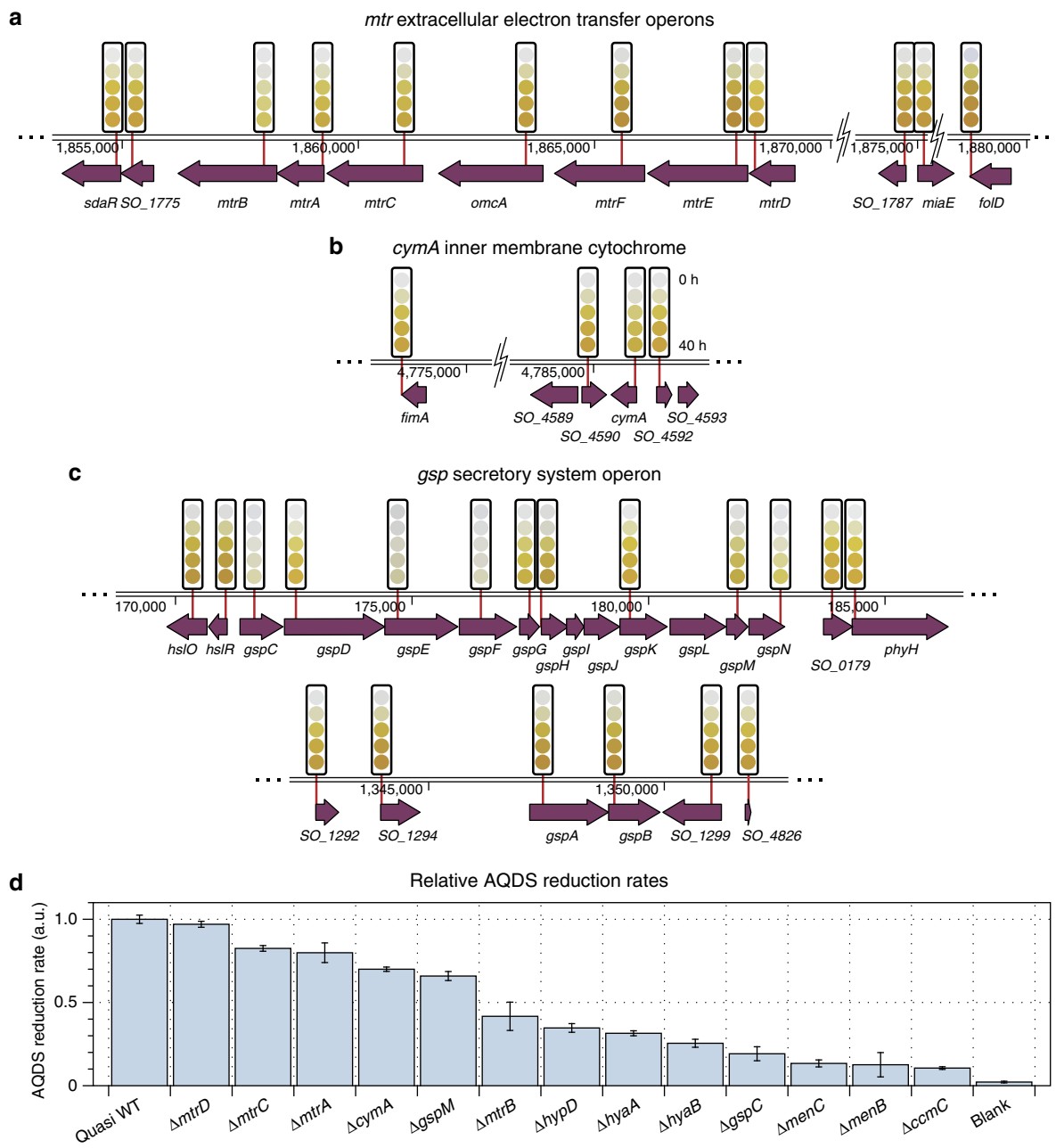

**Figure 6 | Gene disruptions that change AQDS reduction rate.** (**a–c**) AQDS reduction screen results showing gene disruptions that produce small but detectable changes in AQDS reduction rate. (**d**) Linear AQDS reduction rates of selected mutants relative to an averaged quasi wild-type (WT) control (Methods). Error bars are the s.d. of three replicates. A full listing of rates for all mutants can be found in Supplementary Data 6.

The high-throughput, high-sensitivity AQDS reduction measurements shown here provide a template for a comprehensive search for mutants that may produce only weak phenotypic changes from the wild-type. In contrast, in conventional measurements these small variations from wild-type behaviour (for example, those produced by disruptions to *mtrA*, *mtrC* and *cymA*) may not be apparent to the unaided eye, especially if the experimenter must sort through an extremely large random collection. These types of comprehensive experiments could give insights into phenotypes produced by genes with multiple overlapping functions such as EET.

## Discussion

To date, the accelerating move of microbial genetics toward non-model organisms has been hindered by the scarcity of low-cost, high-quality and broadly applicable genetic tools. Whole-genome knockout collections are among the most valuable tools for any organism. However, the high cost of constructing these collections by targeted gene deletion methods like those used to build YKO and the Keio Collection[39] put them out of reach for most investigators. Many of the features of these gold standard collections can be reproduced in collections made by the cataloging and condensation of extremely large transposon mutant collections. While it is fair to say that gene disruption

by transposon mutagenesis is not strictly equivalent to gene deletion, it is sufficiently similar to be useful. In particular, a common concern with transposon mutagenesis is the possibility of prevention of transcription of downstream genes, particularly in an operon, interfering with a reliable connection of genotype to phenotype. However, these polarity effects can be mitigated by the use of carefully chosen transposon sequences[23,42] such as the modified pMiniHimar sequence used here that lack transcriptional terminator sequences[60].

While recent advances in robotic combinatorial pooling have significantly lowered the cost of cataloging large transposon insertion mutant collections, and allow for robust sequence analysis and mapping, the hardware needed is not yet widely deployable and its costs still pose a high barrier to adoption. While simple manual systems are cheaper, they do not offer the same type of robust reconstruction due to complications with sample multiplicity.

We have developed a probabilistic reconstruction algorithm that allows the use of much simpler, faster and lower-cost pooling than established methods. This facilitates an up to 100-fold increase in speed and at least a 20-fold cost reduction relative to the next cheapest comparable method.

The reconstruction algorithm forms the centre of Knockout Sudoku, a complete system to enable rapid cataloging of extremely large transposon mutant collections, automatic selection of a set of representative mutants, followed by validation and curation to produce a QC collection. This method is generally applicable to any microorganism into which a transposon can be introduced. We have used this method to condense a $\approx 40,000$ member progenitor collection of *S. oneidensis* mutants to a QC collection with representatives for 3,667 genes. The QC *S. oneidensis* Sudoku collection was assembled with a dedicated team of two, for a budget of only approximately \$12,000 (Supplementary Table 1). Knockout Sudoku opens the possibility of rapidly and cheaply constructing whole-genome knockout collections for many non-model, even esoteric organisms and possibly strains engineered for specific studies.

The recapitulation and extension of many of the results of earlier studies on AQDS reduction by *S. oneidensis* demonstrate the basic functionality of the *S. oneidensis* Sudoku collection. This gives us confidence that future collections made by Knockout Sudoku can bring the quality of genetic characterization now only possible with tools like the KC and YKO to many more microbial species and cell lines at an affordable cost. As a result, many more investigators can discover genes underlying many fascinating capabilities not seen in model organisms.

## Methods

**Simulation of transposon mutant collection generation.** The appropriate size of the progenitor collection was estimated by two methods: an analytical model based upon Poisson statistics that is a function solely of non-essential gene count, $N$, and a numerical Monte Carlo algorithm (Supplementary Fig. 1, algorithm 1) that combines the published genome sequence of *S. oneidensis* with previously published gene essentiality information.

The Monte Carlo algorithm generates a list of AT and TA dinucleotide[52] positions that are suitable for transposon insertion in the *S. oneidensis* genome (GenBank accession numbers: chromosome: NC_004347.2; megaplasmid: NC_004349.1). The algorithm builds a lookup table indexed by possible insertion location, records the locus name in which this position falls and marks the location as either 'Dispensable', 'SurpriseDispensable', 'GuessDispensable', 'DispensableUnclearInEcoli', 'ExpectedEssential', 'NewEssential' or 'Unknown' using data from Deutschbauer *et al.*[13]. Possible insertion locations that did not fall into any locus were assigned to a non-coding locus and marked as 'Dispensable'. We considered two scenarios: a low non-essential count, $N$, case where the disruptable genes were marked as 'Dispensable'; 'SurpriseDispensable'; 'GuessDispensable' and 'DispensableUnclearInEcoli', and a high non-essential count case where all genes marked 'Unknown' were also considered non-essential. The low non-essential count case considered 3,256 genes while the high case considered 4,184 genes.

Picking events were simulated by a random choice (with replacement) of a non-essential transposon insertion location. If the hit count for that locus was previously zero, the unique gene hit count, $n_{insertions}$, was incremented by 1. In total, we computed 1,000 trials of 46,708 picks for both the high and low non-essential count scenarios and used these to compute the average value and standard deviation of $n_{insertions}$ at each pick, $k$ (Fig. 2c, solid lines).

The size of a transposon insertion collection necessary to achieve adequate coverage of a microbial genome can also be calculated through an analytical model by assuming a Poisson distribution of transposon insertions. Assuming a genome with $N$ non-essential genes, and a collection of $k$ transposon insertion mutants, a given gene should see $x$ insertions with a probability,

$$P(x, k, N) = \exp(-k/N)(k/N)^x(1/x!). \qquad (1)$$

The probability that a single gene will see no transposon insertions,

$$P(0, k, N) = \exp(-k/N). \qquad (2)$$

By linearity of expectation, the total number of genes that will see no insertions,

$$n_{no\ insertions} = N \exp(-k/N). \qquad (3)$$

Or, the total number of genes for which insertions will be found,

$$n_{insertions} = N(1 - \exp(-k/N)). \qquad (4)$$

The analytically derived value of $n_{insertions}$ is plotted for the high and low non-essential gene count scenarios in Fig. 2b as dashed lines.

The simulated rarefaction curves were compared against an experimentally derived curve using random sampling (without replacement) from the 46,708 non-unique transposon insertion instances in the progenitor collection catalogue. This process was repeated 1,000 times to generate a standard deviation around the mean of the rarefaction curve (Fig. 2d, solid black lines).

A program that creates an index linking transposon insertion location to genomic locus (KOSUDOKU-EGENEINDEX) and a program that implements the Monte Carlo simulation of transposon insertion (KOSUDOKU-COLLECTIONMC) are included in the KOSUDOKU package.

**Preparation of progenitor collection.** We constructed a transposon insertion plasmid based on pMiniHimar[21] containing flp-recombinase sites to permit later excision of the antibiotic resistance cassette (Supplementary Data 1). The progenitor transposon insertion collection was prepared by delivery of the modified pMiniHimar plasmid to *S. oneidensis* by conjugation with *E. coli* WM3064 (ref. 61). The transposon insertion library was plated onto LB agar Petri dishes with 30 µg ml$^{-1}$ kanamycin at a density of $\approx 200$ colonies per plate. In total, 39,198 colonies were picked into the wells of 417 96-well polypropylene plates with two sterile control wells per plate with a CP7200 colony picking robot (Norgren Systems, Ronceverte WV, USA). Each well contained 100 µl per well of LB with 30 µg ml$^{-1}$ kanamycin. Plates were sealed with a sterile porous membrane (Aeraseal, Catalog Number BS-25, Excel Scientific) and incubated at 30 °C with shaking at 900 r.p.m. until saturation was reached (typically $\approx 21$ h later).

The total progenitor collection size was decided upon by considering the results of the simulation of represented gene count (Methods) and our estimate of the maximum size of collection that we could pick, incubate and pool before risking perishment of any part of it.

**Progenitor collection combinatorial pooling and preservation.** The progenitor collection was pooled shortly after completion of growth using a 96-channel pipettor (Liquidator-96, Rainin) and specialty SBS format row (1 used) and column plates (1 used) (Catalog Numbers RRI-3028-ST and RRI-3029-ST, Phenix Research Products, Candler NC, USA) for row and column pool preparation and single-well trays (OmniTray, Catalog Number 242811, Nalgene Nunc) for plate row (20 used) and plate column pool (21 used) preparation.

For each plate in the progenitor collection, $4 \times 10$ µl aliquots were withdrawn from each well with the 96-channel pipettor and dispatched to the row plate for row pooling; column plate for column pooling; a single-well plate designated as the plate-row pool receptacle for the plate; and another single-well plate designated as the plate-column pool receptacle for the plate. All four operations were completed in approximately 2 min. In total, the entire progenitor collection was pooled with $4 \times 417 = 1,668$ operations. Following pooling the plate was covered with a sterile lid and set aside until all plates were pooled.

Upon completion of pooling, all plates were cryopreserved by the addition of 50 µl (approximately equal to the remaining volume of culture after withdrawal of pooling aliquots and evaporation) of sterile 20% glycerol. Glycerol was mixed by shaking at 900 r.p.m. and plates were immediately frozen at $-80$ °C. Pooling and cryopreservation was completed in a single day ($\approx 18$ h) with a team of five people.

**Pool amplicon library generation and sequencing.** Amplicon sequencing libraries were generated from the mutant pools through a two-step hemi-nested PCR reaction[62,63] (Supplementary Fig. 2). The reaction amplified a portion of the chromosome adjacent to the transposon present in each collection member and added Illumina TruSeq flow-cell-binding and read-primer-binding sequences to the 3'- and 5'-ends of the amplicon while replacing the standard Illumina index sequence with a custom barcode sequence for each pool. Genomic DNA

was extracted from the coordinate pools with a genomic DNA extraction kit (Quick gDNA-MiniPrep, D3006, Zymo Research).

The first PCR step used a total reaction volume of 20 μl per well with two units of OneTaq DNA Polymerase (M0480, New England Biolabs, Ipswich MA, USA) and 200 nM of each primer (Integrated DNA Technologies, Coralville IA, USA), and 200 μM of dNTPs (N0447, New England Biolabs). The first reaction was templated with 2 μl per well of purified genomic DNA at a concentration of ≈100 ng μl$^{-1}$, and used the three primers HimarSeq1.2, CEKG2C-IllR2, and CEKG2D-IllR2 (Supplementary Data 5) in the following reaction cycle: first 5 min at 95 °C; 6 cycles of 30 s at 95 °C, 30 s at 42 °C (lowering by 1 °C per cycle), 3 min at 68 °C; then 24 cycles of 30 s at 95 °C, 30 s at 45 °C, 3 min at 68 °C; and finally 5 min at 68 °C followed by storage at 4 °C. Note that we do not attempt to normalize the amount of template loaded into the reaction for the amount of each species estimated to be in each pool.

Each second step PCR reaction used a total volume of 50 μl with 5 units of OneTaq polymerase and was templated with 0.5 μl of the completed corresponding first PCR reaction. The second step used four forward primers at 50 nM each (HimarSeq2.4–4N, HimarSeq2.4–5N, HimarSeq2.4–6N and HimarSeq2.4–7N). Each of these primers contains between four and seven random bases in order to avoid overloading a single colour channel in the Illumina sequencer imaging system. Each amplicon pool used a different barcoding primer at 200 nM (Supplementary Data 5). This step used the reaction cycle: 30 cycles of 30 s at 95 °C, 30 s at 56 °C, 3 min at 68 °C; 5 min at 68 °C; and finally storage at 4 °C.

Products of this final reaction were pooled and were purified by agarose gel electrophoresis followed by gel extraction (Gel DNA Recovery Kit, Catalog Number D4001, Zymo Research) of the section of the library with molecular weights between 500 and 1,000 bp. The libraries were combined and single-end sequenced from the transposon side to at least 40 bases past the junction on two lanes of an Illumina HiSeq 2500 in Rapid Run mode with 67 bp single-end reads.

**Analysis of sequencing data set.** We used a suite of custom algorithms developed in PYTHON with the SCIPY[64,65] and NUMPY[66] libraries to condense the large volume of sequencing data into a collection address catalogue (Fig. 3 and Supplementary Fig. 1). Each read in the sequencing data set was examined by a regular expression that matched the main sequence (Illumina read 1) with that of the transposon and the index sequence (Illumina read 2) to a pool barcode with up to 2 mismatches allowed in each case (Supplementary Fig. 1, algorithm 2, KOSUDOKU-SEQANALYZE). The genome sequence portions of those reads with a valid barcode and transposon sequence were aligned against the S. oneidensis MR-1 chromosome and megaplasmid sequences (GenBank ID accession numbers NC_004347.2 and NC_004349.1) using BOWTIE2 (ref. 67)) in end-to-end alignment mode to yield a genomic transposon insertion coordinate. This data was condensed into a pool presence table that enumerates the number of reads corresponding to each pool for each transposon insertion coordinate, giving a set of coordinates that map back to the progenitor collection (Fig. 3b).

Each pool presence table line is thresholded to remove collection coordinates with low read counts. The read count threshold is scanned from 1 to 30 to determine a solution to the pool presence table in which the number of lines that map to single addresses is maximized while maintaining a large number of lines that map in any way and stabilizing the number that do not map. We found that a read count threshold of five produced the most satisfactory compromise between these requirements (Supplementary Fig. 1, algorithm 3, KOSUDOKU-POOLANALYZE; Supplementary Fig. 3).

Possible progenitor collection addresses were calculated for each line in the pool presence table by generating all possible combinations of coordinates with a read count above a predefined threshold. Lines with only one entry per coordinate axis mapped unambiguously to a single line, while lines with multiple coordinates in only one axis mapped unambiguously to multiple lines.

A significant minority of lines map ambiguously to multiple addresses. An example of this is shown in line 3 of Fig. 3b. Intuitively, one would expect that if a given mutant in a given well produced a large number of reads in one coordinate axis, it would also do so in the other three. Thus, we expect that the two correct addresses for this line are composed of just high and just low sets of read count coordinates.

Pool presence table lines (transposon insertion coordinates) that map to single collection addresses are used to calculate the six ratios of reads between the pool axes (row/column; row/plate-column; row/plate row; column/plate-column; column/plate-row; plate row/plate-column). The natural logarithms of these ratios are used to generate a set of 6 histograms and are fit with a Voigt function (Fig. 3c–h and Supplementary Fig. 1, algorithm 4, KOSUDOKU-POOLFIT). The read number ratio fits were integrated and normalized to an area of 1.0 to generate a set of six probability distribution functions.

The read count ratios are calculated for each possible progenitor collection address that can be generated from the address coordinates in an ambiguously mapping pool presence table line. The probability of this read count ratio is assessed by using the probability distribution function for that ratio, and the total probability of the proposed address is calculated by multiplying these six probabilities together (Supplementary Fig. 1, algorithm 5, KOSUDOKU-POOLSOLVE). The most probable addresses, up to a maximum defined by the highest number of coordinates in a single axis for that line, are mapped to the collection catalogue.

A small number of lines had coordinates in only 3 axes that were above threshold, but did have a coordinates in the fourth axis. In these cases, the highest count coordinate was taken from the fourth axis to make a complete set of coordinates and allow mapping.

A small number of transposon insertion locations produce amplicons that align to multiple consecutive coordinates in the S. oneidensis genome. These apparently consecutive mutants were grouped following mapping to locations in the progenitor collection. If the run of consecutive coordinates contains an AT or TA dinucleotide, the A of this pair is taken as the consensus coordinate. Otherwise, the sum of all reads used to map the mutants is used to calculate a median consensus coordinate.

**Sanger verification of location inference predictions.** Mutant location inference predictions were verified by a two-step semi-nested PCR reaction[62,63] similar to that used for pool amplicon library generation (Methods). The reaction amplified a portion of the chromosome adjacent to the transposon in the mutant.

The first PCR step used a total reaction volume of 20 μl per well with two units of OneTaq DNA Polymerase (M0480, New England Biolabs), 200 nM of each primer (Integrated DNA Technologies), and 200 μM of dNTPs (N0447, New England Biolabs). The first reaction was templated with 1 μl per well of saturated bacterial culture, and used the three primers HimarSeq2, CEKG2C and CEKG2D (Supplementary Data 5) in the following reaction cycle: first 5 min at 95 °C; 6 cycles of 30 s at 95 °C, 30 s at 42 °C (lowering by 1 °C per cycle), 3 min at 68 °C; then 24 cycles of 30 s at 95 °C, 30 s at 45 °C, 3 min at 68 °C; and finally 5 min at 68 °C followed by storage at 4 °C.

Each second step PCR reaction used a total volume of 20 μl with two units of OneTaq polymerase, 200 nM of the primers HimarFRTSeq2 and CEKG4 (Supplementary Data 5) and was templated with 0.5 μl of the completed corresponding first PCR reaction. The following reaction cycle was used for the second step: 30 cycles of 30 s at 95 °C, 30 s at 56 °C, 3 min at 68 °C; and finally 5 min at 68 °C followed by storage at 4 °C.

The second step PCR reaction was purified by a standard PCR clean-up procedure (Genewiz, South Plainsfield NJ, USA) and sequenced by Sanger sequencing (Genewiz, South Plainsfield NJ, USA).

The measured transposon insertion location was found by alignment of the Sanger read against the S. oneidensis genome sequence with BLAST[68], and compared against the predicted transposon insertion location with a custom program, KOSUDOKU-VERIFY, developed with PYTHON and the BIOPYTHON library[69]. KOSUDOKU-VERIFY is included in the KOSUDOKU package. A complete set of results is shown in Supplementary Data 3.

**Selection of mutants for quality-controlled collection.** The progenitor collection catalogue (Supplementary Data 2) was used to assemble a quality-controlled non-redundant collection of mutants containing representatives for 3,282 genes disrupted in the progenitor collection.

We developed a set of algorithms to choose a representative mutant for each gene disrupted in the progenitor collection. Mutants were chosen for inclusion in the condensed collection by a scoring function that estimated the likelihood that a randomly picked colony struck out from a well containing a representative disruption of a gene would disrupt the function of that gene (Supplementary Fig. 6, algorithm 6, KOSUDOKU-CONDENSE). The function first assessed the probability that the colony would be the desired mutant given the level of co-occupancy of the well. Then, given that the right colony was picked, the function estimated the probability that the disruption would knock out the function of the gene as the fractional distance of the transposon from its translation start.

We re-arrayed 2,699 mutants from singly occupied wells and colony purified representatives for another 999 loci from multiply occupied wells. For the mutants from multiply occupied wells, we streaked a sample from each well and picked between 2 and 10 colonies, which we estimated would give an 85% chance of finding the desired mutant, into the wells of a series of 96-well plates. In total, this collection comprised 5,653 samples, arrayed on 84 plates. The collection was arranged into a 10 × 9 plate grid, re-pooled and resequenced to assist in sorting well co-occupants from desired mutants (Methods).

**Orthogonal verification of quality-controlled collection.** The sequencing data from the quality-controlled collection (Methods) was analysed through an orthogonal method to test the sequence contents of each well against the predicted contents, rather than locate each sequence (Supplementary Fig. 6, algorithm 7, KOSUDOKU-ISITINTHERE). The verification algorithm generates a list of all transposon coordinates appearing in each coordinate pool. The sequence content of all wells in the quality-controlled collection can be determined by calculating the intersection of the four transposon coordinate sets that correspond to the four pool coordinates of the well. If the intersection of the four pools contained one of the predicted genomic coordinates for that well, it was marked as correct. If the intersection contained the coordinate that we hoped to isolate by colony purification, it was also marked containing a desired mutant. This program generates a provisional condensed collection catalogue. The set-intersection algorithm was implemented in PYTHON and the SCIPY[64,65] and NUMPY[66] scientific computing libraries.

***Shewanella* basal media.** Our AQDS reduction assay experiments used minimal media consisting of (per litre): ammonium chloride ($NH_4Cl$) (0.46 grams); dibasic potassium phosphate ($K_2HPO_4$) (0.225 g); monobasic potassium phosphate ($KH_2PO_4$) (0.225 g); magnesium sulfate ($MgSO_4.7H_2O$) (0.117 g); and ammonium sulfate (($NH_4)_2SO_4$) (0.225 g). The media was buffered with 100 mM HEPES its pH was adjusted to 7.2.

**High-throughput assay for AQDS reduction activity.** To functionally validate the quality-controlled collection, we screened it for reduction of the redox dye AQDS[55]. Briefly, the humic substance analogue AQDS changes colour from clear when oxidized to deep orange when reduced by two electrons (Fig. 5a).

A frozen stock of the quality-controlled S. oneidensis Sudoku collection was duplicated with a multi-blot replicator (Catalog Number VP 407, V&P Scientific, San Diego CA, USA) into 96-well plates containing 100 µl of LB media with 30 µg ml$^{-1}$ kanamycin per well to minimize contamination from external sources. The plates were sealed with an air porous membrane (Aeraseal, Catalog Number BS-25, Excel Scientific) and grown to saturation overnight (typically 21 h) at 30 °C with shaking at 900 r.p.m. The following day the plates were imaged with a photographic data acquisition system (the macroscope). An image analysis algorithm developed with SciKit Image[70] and SimpleCV[71] tested the plate images for cross-contamination and growth-failure events by comparison with the collection catalogue. The image analysis algorithm updated the record for each well in the collection catalogue with growth information to assist in rejection of false positives in the EET screen due to growth failure.

A 10 µl aliquot of culture from each well was transferred to a barcoded clear 96-well assay plate in which each well contained a mixture of 70 µl of buffered *Shewanella* Basal Media (Methods) and 20 µl of 25 mM AQDS, for a final concentration of 5 mM. The assay plates were immediately transferred to a vinyl anaerobic chamber (Coy Laboratory Products, Grass Lake MI, USA) with an atmospheric $H_2$ concentration of ≈3%. For each plate in the collection, we took three copies of every mutant and imaged them multiple times over the course of approximately 60 h with a macroscope device inside the anaerobic chamber. Hits were identified by visual inspection and an image analysis algorithm (Methods).

Hits were validated by colony purification and selection of three colonies into the wells of a 96-well plate. Retest plates were grown to saturation overnight and tested again. Each retest plate contained disruption mutants for genes flanking the candidate hits in the S. oneidensis genome that acted as quasi wild-type experimental controls and calibrants in image analysis. A representative sample was picked for each hit and its transposon identity was confirmed by Sanger sequencing.

**Image analysis.** A custom image analysis program developed with SCIKIT IMAGE[70], SIMPLECV[71] and MATPLOTLIB[72] was used to sort the AQDS reduction assay images by barcode and identify well positions and assign well content information. Almost all information on the reduction state of the AQDS dye can be found in the blue colour channel of the assay plate images. At the start of the assay, all colour channels are saturated (resulting in a set of white wells). As the AQDS dye is reduced and becomes orange, the intensity of the red channel remains approximately constant, with a small reduction in green channel intensity and a large drop in the blue channel. Time courses of the reduction of the AQDS dye for each mutant (Supplementary Fig. 7A,B) were generated by calculating the mean blue channel intensity for all pixels at the centre of each well in each image in the time series. Mutants that had grown successfully to saturation but displayed no significant change in blue channel intensity were reported as candidate hits.

The time series of colours for each gene shown as coloured circles above each gene in and Supplementary Figs 7 and 8 were generated by an algorithm that interpolated the multi-replicate average of mean well-centre colour values for that mutant at 0, 10, 20 and 39.5 h (the length of the shortest time series in all experiments) after the initiation of the reduction experiment. All points in Figs 5 and 6 and Supplementary Figs 7 and 8 were calculated from the mean of three data points while error bars are the s.d. of that average.

Rates of reduction were calculated by a linear fit to the linear portion of the blue-channel reduction curve with DATAGRAPH (Visual Data Tools). The reduction rate in units of moles per hour was computed using a conversion factor derived from the quasi wild-type controls. Results are shown in Supplementary Data 6.

**Parental strain source.** The parental organism for the S. oneidensis Sudoku Collection was provided by Professor Jeffrey Gralnick (University of Minnesota). The identity of this organism was verified by whole-genome sequencing and comparison with the archived S. oneidensis genome (GenBank accession codes NC_004347.2 and NC_004349.1).

**Software availability.** The KOSUDOKU package and sample input and output data are available at https://github.com/buzbarstow/kosudoku.

**Data availability.** Next-generation sequencing data sets are available the NCBI Short Read Archive under accession codes SRX1837558 and SRX1837561. AQDS reduction image data is available upon request from the authors. The sequence of

the modified pMiniHimar plasmid is included as Supplementary Data 1. The authors declare that all other data supporting the findings of this study are available within the article and its Supplementary Information files or are available from the corresponding author upon request.

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

## Acknowledgements

We thank N. Ando, K. Davis, A. Palmer, S. Meisburger, B. Chang, E. Adler, K. Malzbender and C. Kyauk for experimental assistance; W. Metcalf for providing *E. coli* strain WM3064; J. Gralnick for providing *Shewanella oneidensis* MR-1; L. Kovacs, J. Miller, L.R. Parsons, S. Silverman, W. Wang and J. Wiggins for assistance with next-generation sequencing and media preparation; N. Ando, B.L. Chin, L. Kugelmass, P.A. Silver and J. Way for critical reading of this manuscript. This work was supported by a Career Award at the Scientific Interface from the Burroughs Wellcome Fund and Princeton University startup funds (B.B.).

## Author contributions

B.B. and M.B. conceptualized and developed the Knockout Sudoku method and the *S. oneidensis* knockout collection. B.B., M.B., L.S., I.A.A and O.A. designed and performed experiments. B.B. developed and implemented the analysis software. B.B. and M.B. wrote this article. All authors reviewed and revised the manuscript.

## Additional information

**Competing financial interests:** M.B., L.S., I.A., O.A. and B.B are seeking patent protection for the probabilistic reconstruction algorithm used in the Knockout Sudoku method.

