## [Peer Review File · Nature Communications]

Reviewer #1 (Remarks to the Author)

Baym et al. present a new method to rapidly develop an ordered transposon library using *Shewanella oneidensis*. The method itself is agnostic and should be rapidly applicable to many different organisms where such a tool would be of use. Given this applicability, many people will find the method quite useful and is suitable for Nature Communications. My only complaint with the paper relates to their findings on AQDS reduction pathways. First, because AQDS can easily get into the periplasm and also likely crosses the cytoplasmic membrane, it is not a good model for extracellular electron transfer (EET). Moreover, the authors do not draw any distinction between cells that cannot grow under their test condition (in the anaerobic chamber) and cells that cannot reduce AQDS. The biological inferences are somewhat tangential to the focus of the paper and should only require revisions to the text.

Major Comments

Polarity is not mentioned in this manuscript, even though transposon mutations almost always have this effect. Still need to generate clean deletions to obtain clear genetic evidence to tie a phenotype to a gene, or to robustly complement the defect.

The AQDS reduction assay should not be described as 'EET activity' or 'EET screen'. AQDS is not 'an analog for humic substances,' rather it is an analog for redox active moieties that can be found in humic substances. AQDS has been shown to enhance extracellular electron transfer to minerals by dissimilatory metal reducing bacteria, but it is not a good model for 'EET' because the compound is known to enter cells (which is why *tolC* mutants are critical to survival). If AQDS reduction is occurring inside the cell, it should not be referred to as 'extracellular electron transfer.' In the Functional Validation section (beginning on line 205) the authors should differentiate between mutations that are expected to have a specific defect in reducing AQDS and mutations that prevent the cells from growing under the conditions tested. Instead of arguing for the 'high apparent replaceability' of *MtrA*, *MtrC* and *CymA* (line 246) the authors should consider that the majority of reduction could be occurring inside cells and is not dependent on EET. To be clear, inside could be in the periplasm or in the cytoplasm. One final point about this, how do the authors know that their mutations in *mtrA*, *mtrC* or *cymA* are 'good' and have the expected phenotype? Mutants lacking any of these genes should be impaired in iron reduction. Mutants defective in *cymA* should lose the ability to grow on a plethora of electron acceptors (except oxygen, TMAO and thiosulfate). If the authors wish to make statements about the role (or lack thereof) of these genes, they'll need to be phenotypically validated.

The authors should discuss the number of mutations in their library relative the number of known non-essential genes in *S. oneidensis* as reported in references 12, 25 and in Yang et al., 2014, PLoS Comp Biol.

Line 37 - *MtrB* is not a cytochrome.

Line 39 - missing the most recent (and one of the most important) contribution to the debate: Sturm et al., 2015, ISME J 9(8):1802-11.

Line 42 - this paragraph should (perhaps) point out that most transposon mutagenesis experiments are created explicitly for a specific screen. Subsequent screens are performed on a new pool of mutants. In some organisms, the ease of generating the transposon mutant pool obviates the need for a curated, preserved library.

Line 209 - the electron donor for AQDS reduction should be mentioned here. The authors speculate about the electron donor on line 223. It is unclear why lactate was not added, considering that this is the substrate typically used when studying EET in *S. oneidensis*.

Line 227 - also consistent with AQDS reduction not being 100% dependent on EET, especially since there is evidence AQDS can get into cells (reference 51), where it could be reduced by a number of different mechanisms (as discussed above).

Minor Comments

Why 'sudoku'? Is there a mathematical tie in to the number-game? Just curious (a reader might be as well).

Line 26 - unclear what the authors mean by 'metal deposits'

Line 28 - This claim related to 'largest complements of cytochrome' is not easily made, considering essentially all of the sequenced *Geobacter* strains have 2-3 times as many predicted cytochromes'. It should also be specifically noted that the authors are referring to c-type cytochromes.

Line 50 - unclear why reference 32 is included here. Also reference 31 was later demonstrated to not be 'EET', rather cross-complementation: Myers and Myers, 2004, AEM, 70(9):5415-25

Line 75 - what is 'suitably large' here?

Line 82 - why not be specific about what the assay tests for?

Line 86 - is *Shewanella* really 'esoteric'? There are over 2200 references in PubMed.

Line 176 - why should there be twice as many insertions per 10,000 bp on the megaplasmid?

Line 189 - Why was the 'representative' insertion in *mtrD* chosen to be very close to the end of the gene? Insertions near the end of the gene can sometimes result in a lesser, or no, phenotype. The fold insertion shown in Figure 4 is very likely in this category as folate biosynthesis should be essential to the cells. It should also be noted that insertions at the very start of a gene can also result in an altered phenotype. Large Tn-seq datasets can be used to quantify these (see Yang et al., 2014).

Line 201 - perhaps mention here how much effort it was to later add 385 mutants?

Line 203 - be cautious about using the term 'disruptable loci' - for example, insertions in *feoA* and *feoB* are possible, but absent from your collection (based on figure 4).

Line 679 - why was kanamycin added? The transposon cannot move without the transposase encoded on the plasmid (which cannot replicate in *S. oneidensis*).

Italics missing throughout the references.

Reviewer #2 (Remarks to the Author)

The combination of transposon mutagenesis with next generation sequencing has enabled many high-throughput studies in microbiology including those that have assigned new functions to (un)known genes, the discovery of non-coding (regulatory) RNAs and the association of genes and pathways with drug sensitivity and disease induction.

In this manuscript the authors describe a relatively straightforward and cheap method to construct an ordered knockout array from a pooled library of random transposon mutants. Even though experiments with pooled libraries of transposon insertion mutants are often high-throughput, experiments that validate the obtained genotype-phenotype interactions require experiments in

which specific KO mutants are used. Since making KO mutants may be labor intensive, such validations are often omitted or done at a very limited scale. Additionally, experiments based on pooled libraries can for instance mask the existence of 'cheaters', i.e. mutants that benefit from the genotype of others in the population but may not be able to grow efficiently by themselves. The construction of ordered knockout arrays can thus, without a doubt, serve as a great resource.

However, the major issue I have with this manuscript is that it is unclear how the presented method differs from the combinatorial approach that was published by Goodman et al., 2009 in *Cell, Host & Microbe*. In that paper the researchers not only present a novel method named INseq to perform pooled transposon experiments, they also present an original combinatorial approach to construct an ordered array of transposon mutants. After reading the current manuscript, I fail to see the difference with the original method or a significant improvement upon it.

Besides that I have two additional comments:

1. I could not find a clear explanation of how the scripts that assign genome insertion locations to 96-well plate coordinates exactly work. I also couldn't find where the scripts can be downloaded from (e.g. GitHub), and whether there is a manual that describes how to 'operate' the scripts. Since the authors are presenting the method as a novel and complete-package that can be performed by basically any microbiology lab, I believe there should be a detailed analyses package associated with the manuscript that enables an essential part of this method.
2. I found the section from lines 143 to 182 not particularly clear. After reading it three times I still wasn't sure how many mutants were mapped, what the specific issues were with mutants that couldn't be mapped right away, how this was solved and what the final tally was of mapped vs unmapped mutants.

Reviewer #3 (Remarks to the Author)

Main problem to review this work was that documents were incomplete and they did not include any online methods (where everything was hopefully much better explained). Seems that in most places on the manuscript they explain very little and simply referred to online methods. I can still comment something in the methodological scheme (based on the information provided in the actual manuscript):

1. Transposon insertions are assumed to be Poisson-distributed (the authors should comment how strong is the assumption that insertions occur independently, which follows directly from Poisson distribution assumption). The authors simply use Poisson probability formula to obtain lower estimate of non-essential gene set.
2. Monte Carlo simulations are used to obtain disrupted gene count by random draws from the observed set of transposon insertion mutants. They used the most conservative of the two estimates (one given by Poisson formula and one given by MC simulation). Authors should comment why MC was giving larger estimate than Poisson distribution. Explain also what is the rationale of using MC sampling, why it can work here?
Explain also Figure 2 much better.
2. Bayesian inference is mention in the abstract but the actual use of Bayesian methods is limited to the use of simple Bayes formula if I understood right (based on minimal explanation provided in the ms), not any complicated MCMC calculations or fancy models. Therefore, "Bayesian inference" in the abstract may be good to replace with "Bayes formula".
Please provide also better explanation of that in the actual manuscript. Currently I was not able to obtain idea of the whole story based on the current text.

3. The part before application of Bayes formula, some numbers to be substituted to formula were given by fit of Voigt function. To me this looked like approximating shape of empirical distribution of something with this function.

That parts was very interesting but again in the manuscript there was completely insufficient explanation of the procedure (hopefully online methods did the better job). Please add some explanation also to the manuscript part.

With very limited explanations I was not able to understand much more from statistical methods applied. I hope this can still be of some help.

Response to Reviewer 1

In summary, reviewer 1 comments:

“Baym et al. present a new method to rapidly develop an ordered transposon library using Shewanella oneidensis. The method itself is agnostic and should be rapidly applicable to many different organisms where such a tool would be of use. Given this applicability, many people will find the method quite useful and is suitable for Nature Communications.”

The main concern of reviewer 1 is that the AQDS reduction assay presented in the manuscript does not serve as the best currently available model for extracellular electron transfer (EET):

“My only complaint with the paper relates to their findings on AQDS reduction pathways. First, because AQDS can easily get into the periplasm and also likely crosses the cytoplasmic membrane, it is not a good model for extracellular electron transfer (EET).”

However, reviewer 1 notes that:

“The biological inferences are somewhat tangential to the focus of the paper and should only require revisions to the text.”

Accordingly, we have modified the text of our manuscript to narrow the scope of our statements regarding EET, while still highlighting the reproduction of many earlier studies on AQDS reduction by *S. oneidensis*. In particular, we have modified statements in the abstract (line 20), and have separated new observations from the Functional Validation section of the Results section (lines 347-373).

Response to Major Comments by Reviewer 1

1. *“Polarity is not mentioned in this manuscript, even though transposon mutations almost always have this effect. Still need to generate clean deletions to obtain clear genetic evidence to tie a phenotype to a gene, or to robustly complement the defect.”*

We very much agree with reviewer 1 that validation of a novel gene to phenotype association (particularly if that gene is part of an operon) will require either a precise targeted deletion of the gene or complementation. To address this concern, we have added a short section to the Discussion regarding polarity, strategies taken to mitigate its effects in the *S. oneidensis* Sudoku Collection and the validation of any genotype-phenotype association highlighted by collections made with this method (lines 380-387) and additional statements to the Results section (lines 359-361).

2. *“The AQDS reduction assay should not be described as 'EET activity' or 'EET screen'. AQDS is not 'an analog for humic substances,' rather it is an analog for redox active moieties that can be found in humic substances. AQDS has been shown to enhance extracellular electron transfer to minerals by dissimilatory metal reducing bacteria, but it is not a good model for 'EET' because the compound is known to enter cells (which is why tolC mutants are critical to survival). If AQDS reduction is occurring inside the cell, it should not be referred to as 'extracellular electron transfer.' In the Functional Validation section (beginning on line 205) the authors should differentiate between mutations that are expected to have a specific defect in reducing AQDS and mutations that prevent the cells from growing under the conditions tested.”*

This is an excellent point, and we take it very seriously. While there is a wealth of data from genetic screens using AQDS reduction dating back over the past 16 years that provides an extensive resource for validation of the *S. oneidensis* Sudoku collection, we do agree that in recent years AQDS reduction has been superseded in genetic screens for EET activity, and the two phenomena should not be considered equivalent. Accordingly, we have updated our abstract to note that the screen used in this manuscript is not for EET activity but for AQDS reduction (line 20 and lines 91-94).

Additionally, we have not only distinguished mutations that have a defect in reducing AQDS from mutations that prevent growth but have also separated the recapitulation of earlier findings (lines 318-346) from additional findings (lines 347-373) in the Results section. Finally, we have noted that AQDS is an analog for the redox active moieties found in humic substances that can act as electron acceptors during anaerobic respiration on lines 320-321 and lines 91-94.

3. *“Instead of arguing for the 'high apparent replaceability' of MtrA, MtrC and CymA (line 246) the authors should consider that the majority of reduction could be occurring inside cells and is not dependent on EET. To be clear, inside could be in the periplasm or in the cytoplasm.”*

This point is well taken, and accordingly we have removed this speculation from the manuscript and added sentences to the Results section that stress the strong possibility that significant (if not the majority of) AQDS reduction is occurring inside the cell (lines 356-357).

4. *“One final point about this, how do the authors know that their mutations in mtrA, mtrC or cymA are 'good' and have the expected phenotype? Mutants lacking any of these genes should be impaired in iron reduction. Mutants defective in cymA should lose the ability to grow on a plethora of electron acceptors (except oxygen, TMAO and thiosulfate). If the authors wish to make statements about the role (or lack thereof) of these genes, they'll need to be phenotypically validated.”*

We take this point very seriously and have updated our manuscript. In the absence of independent validation through complete gene knockout, we report our findings from the AQDS reduction screen on *mtrA*, *mtrC* and *cymA* gene disruption mutants (lines 347-373) rather than claiming these results as functional validation of the *S. oneidensis* Knockout Collection. Additionally, we note that these mutations were likely to not have been seen in earlier AQDS reduction screens with conventional transposon collections because the rate reductions they produce are too small to be discernible by unaided observation (lines 369-371).

5. *“The authors should discuss the number of mutations in their library relative to the number of known non-essential genes in S. oneidensis as reported in references 12, 25 and in Yang et al., 2014, PLoS Comp Biol.”*

We have added text to the Results section comparing the number of representative mutations in the *S. oneidensis* Sudoku Collection with the number of non-essential genes in the *S. oneidensis* genome estimated by Deutschbauer *et al.* (originally ref 12, now 13), Brutinel *et al.* (originally ref 25, now 27) and Yang *et al.* (ref 28) and have added our own estimate of the non-essential gene count on LB in lines 298-318.

6. *“Line 37 - MtrB is not a cytochrome.”*

This point is very well taken. We have updated the text of the introduction section to note the centrality of the MtrABC complex (made up of the MtrA and MtrC cytochromes and MtrB porin) in EET (lines 34-35).

7. *“Line 39 - missing the most recent (and one of the most important) contribution to the debate: Sturm et al., 2015, ISME J 9(8):1802-11.”*

To address this concern, we have added a reference to the recent article by Sturm *et al.* on the periplasmic electron transfer network in *S. oneidensis* to line 37 (ref 19) of the updated manuscript.

8. *“Line 42 - this paragraph should (perhaps) point out that most transposon mutagenesis experiments are created explicitly for a specific screen. Subsequent screens are performed on a new pool of mutants. In some organisms, the ease of generating the transposon mutant pool obviates the need for a curated, preserved library.”*

We have added text to the introductory section of our manuscript (lines 51-53) noting that many transposon mutagenesis screens use non-curated collections created specifically for that screen.

9. *“Line 209 - the electron donor for AQDS reduction should be mentioned here. The authors speculate about the electron donor on line 223. It is unclear why lactate was not added, considering that this is the substrate typically used when studying EET in S. oneidensis.”*

While lactate is the most common substrate used when studying EET (and AQDS reduction) in *S. oneidensis*, we had noted in earlier experiments that *S. oneidensis* was capable of reducing AQDS even without the presence of an explicitly added electron donor (other than residual electron donors in the spent LB). We had been curious about this, and felt that a genetic screen with a comprehensive collection could shed light on this. We have added text (lines 324-326) to the manuscript expanding

on our experimental choices.

10. “Line 227 - also consistent with AQDS reduction not being 100% dependent on EET, especially since there is evidence AQDS can get into cells (reference 51), where it could be reduced by a number of different mechanisms (as discussed above).”

We have added text to the manuscript (lines 356-357) noting that considerable reduction of AQDS could occur inside the cytoplasm, and noting the importance of the *tolC* gene (*SO_2917*) to tolerance of AQDS.

Response to Minor Comments by Reviewer 1

1. “Why 'sudoku'? Is there a mathematical tie in to the number-game? Just curious (a reader might be as well).”

When we first envisioned the process for annotating an oversampled transposon insertion collection by next-generation sequencing, the process struck us as very similar to the number placement game and we initially called our project Sudoku Sequencing. While our reconstruction algorithm ended up being significantly different than Sudoku, we still feel the general principle of reconstructing a full collection from combinatorial occupancy constraints is sufficiently salient to keep the name. We have added a short section (lines 65-77) to the introduction on both the general principles of combinatorial pooling (also in response to comments by reviewer 2) and the origin of the name.

2. “Line 26 - unclear what the authors mean by 'metal deposits'”

By metal deposits we were trying to convey the idea that *S. oneidensis* could reduce solid surfaces. We have clarified this sentence in the manuscript (line 24).

3. “Line 28 - This claim related to 'largest complements of cytochrome' is not easily made, considering essentially all of the sequenced *Geobacter* strains have 2-3 times as many predicted cytochromes'. It should also be specifically noted that the authors are referring to c-type cytochromes.”

Our enthusiasm for *Shewanella* may well have gotten the better of us, and this comment is well taken. We have adjusted this statement to indicate that its complement of cytochromes is large, but not by any means the largest (lines 26-27).

4. “Line 50 - unclear why reference 32 is included here. Also reference 31 was later demonstrated to not be 'EET', rather cross-complementation: Myers and Myers, 2004, AEM, 70(9):5415-25”

We have removed references 31 and 32 for clarity and correctness (line 50).

5. “Line 75 - what is 'suitably large' here?”

In this sentence, our aim was to convey the idea that the progenitor collection was large enough that it would contain representative disruption mutants for a very large fraction of the non-essential genes in the *S. oneidensis* genome. We have edited this sentence to improve its clarity (lines 82-85).

6. “Line 82 - why not be specific about what the assay tests for?”

We have edited this text to note that the genetic screen used to functionally validate the *S. oneidensis* Sudoku Collection is an AQDS reduction assay (lines 91-94).

7. “Line 86 - is *Shewanella* really 'esoteric'? There are over 2200 references in PubMed.”

This really depends upon perspective. From the point of view of the electroactive microbes community, *S. oneidensis* is definitely not esoteric, and it reasonable to say that it is a, if not the, model organism for the field. In fact, we would not have constructed the collection around *S. oneidensis* had we not thought this. For instance, we would not have chosen to pilot the Knockout Sudoku method with *Marinobacter subterranei* (1 reference on PubMed), as interesting as we think this organism is.

However, from a broader perspective, one could make the argument that *S. oneidensis* is esoteric. For

example there are 337,825 references for *E. coli* and 114,622 for *S. cerevisiae* in PubMed. The point that we were trying (perhaps unsuccessfully) to make is that the Knockout Sudoku technology is designed to be broadly distributed and to democratize the construction of comprehensive knockout collections. We believe it is sufficiently low cost that a comprehensive knockout collection can now be made for an organism simply because an individual investigator thinks that it is interesting. This means that it is possible to build a collection around an organism that might offer a unique capability that only a small number of people at the time are interested in, and begin to more comprehensively tease apart its genetic basis.

As we do not think such a manifesto is appropriate for a research article, we have adjusted lines 39-40 to not understate the importance of *S. oneidensis* to the research community, yet still convey the need for the development of improved, easier-to-use, lower-cost genetic tools for more esoteric microbes (of the 1 reference in PubMed order) and highlight the broad accessibility of the method (lines 375-379, lines 406-411).

8. *“Line 176 - why should there be twice as many insertions per 10,000 bp on the megaplasmid?”*

This is an excellent observation. We have added a more refined description of the distribution of transposon insertion across the chromosome and megaplasmid to lines 243-245 of the main text. Simply put, the peak transposon insertion density in both the megaplasmid and chromosome are approximately the same. However, the chromosome is sufficiently long that this density noticeably drops with increasing distance from the replication origin, reducing the average density to approximately half of that seen in the megaplasmid.

9. *“Line 189 - Why was the 'representative' insertion in mtrD chosen to be very close to the end of the gene? Insertions near the end of the gene can sometimes result in a lesser, or no, phenotype. The fold insertion shown in Figure 4 is very likely in this category as folate biosynthesis should be essential to the cells. It should also be noted that insertions at the very start of a gene can also result in an altered phenotype. Large Tn-seq datasets can be used to quantify these (see Yang et al., 2014).”*

We have added clarifying statements regarding the choice of mutants in the Knockout Sudoku condensation process (lines 253-261) and the specific *mtrD* disruption mutant chosen to be part of the *S. oneidensis* Sudoku collection (lines 265-265). Additionally, we have considerably expanded our discussion of the types of transposon insertion mutants seen in the progenitor collection, and their effect on any given phenotype (lines 108-120, lines 297-317).

10. *“Line 201 - perhaps mention here how much effort it was to later add 385 mutants?”*

We have added a short sentence to lines 292-293 describing the small amount of streaking and colony picking required to add the additional 385 mutants.

11. *“Line 203 - be cautious about using the term 'disruptable loci' - for example, insertions in feoA and feoB are possible, but absent from your collection (based on figure 4).”*

We have added considerable extra discussion to the introduction and results section of the manuscript detailing the types of insertions likely to be seen and not seen in the progenitor collection (lines 108-120, lines 297-317)

12. *“Line 679 - why was kanamycin added? The transposon cannot move without the transposase encoded on the plasmid (which cannot replicate in S. oneidensis).”*

We added kanamycin here to minimize the chances of contamination from external sources during this large experiment. We have added a short clarifying statement regarding this to lines 829-830 in the Online Methods.

13. *“Italics missing throughout the references.”*

We have added italics at appropriate places in the reference sections in our updated manuscript.

Response to Reviewer 2

In summary, reviewer 2 comments:

“The combination of transposon mutagenesis with next generation sequencing has enabled many high-throughput studies in microbiology including those that have assigned new functions to (un)known genes, the discovery of non-coding (regulatory) RNAs and the association of genes and pathways with drug sensitivity and disease induction.

In this manuscript the authors describe a relatively straightforward and cheap method to construct an ordered knockout array from a pooled library of random transposon mutants. Even though experiments with pooled libraries of transposon insertion mutants are often high-throughput, experiments that validate the obtained genotype-phenotype interactions require experiments in which specific KO mutants are used. Since making KO mutants may be labor intensive, such validations are often omitted or done at a very limited scale. Additionally, experiments based on pooled libraries can for instance mask the existence of 'cheaters', i.e. mutants that benefit from the genotype of others in the population but may not be able to grow efficiently by themselves. The construction of ordered knockout arrays can thus, without a doubt, serve as a great resource.”

However, reviewer 2 notes that;

1. *“However, the major issue I have with this manuscript is that it is unclear how the presented method differs from the combinatorial approach that was published by Goodman et al., 2009 in Cell, Host & Microbe.”*

We take this comment extremely seriously. While Knockout Sudoku and Goodman’s method both use next-generation sequencing to catalog transposon insertion mutant collections, the similarities end there. Simply put, Knockout Sudoku is between 30 and 100 times faster than the Goodman method, costs at least 20 times less to implement and, owing to its reliance on software, is widely deployable.

In our original submission, we included a short history of the development of methods for the production of whole genome knockout collections and stated that while methods such as that developed by Goodman *et al.* “allow for robust sequence analysis and mapping”, they have the limitation that “the hardware needed is not yet widely deployable”. We noted additionally, “the lack of complexity in the combinatorial pools must be compensated by sophisticated algorithms for the disambiguation of sequencing data”.

In response to this comment, we have included several additions to the text that we hope will further highlight and clarify the advances in Knockout Sudoku over existing methods. Simply put, the key advance in Knockout Sudoku is a reconstruction algorithm that removes the need for the sophisticated robotically-actuated, but slow and expensive combinatorial pooling schemes and permits the use of very simple, fast and low-cost combinatorial pooling schemes with very large collections of transposon insertion mutants.

Accordingly, we have edited the abstract (lines 14-15, lines 16-17) to note that Knockout Sudoku is not simply a scheme for annotation of transposon mutant collections but also includes methods for rapid algorithmically-guided condensation of a large mutant collection, validation of this condensed collection and its curation. We have added a similar statement to lines 78-80 of the main text.

To further clarify the advances made in Knockout Sudoku we have included an extended introduction to combinatorial pooling (lines 65-77) and the limitations of robotic combinatorial pooling methods to lines 71-73, and an acknowledgement of the challenges faced by manual methods to lines 75-77. We have added a short sentence comparing the time and cost of combinatorial pooling by a robotic scheme such as that described by Goodman *et al.* and the manual scheme used in Knockout Sudoku (lines 82-87) and have provided additional support for this claim in **Supplementary Note 1**. In addition, we have added short sentences to the Results section to contrast Knockout Sudoku with the Goodman method (lines 158-159). Finally, we have added further clarifying sentences to the Discussion section (lines 388-392 and lines 396-405).

2. *“I could not find a clear explanation of how the scripts that assign genome insertion locations to 96-well plate coordinates exactly work. I also couldn't find where the scripts can be downloaded from (e.g. GitHub), and whether there is a manual that describes how to 'operate' the scripts. Since the*

authors are presenting the method as a novel and complete-package that can be performed by basically any microbiology lab, I believe there should be a detailed analyses package associated with the manuscript that enables an essential part of this method.”

To address this concern, we have made the KOSUDOKU package available on Github. A hyperlink to this repository is included on lines 412-415 of the revised manuscript. In addition, we have updated the flow charts in **Supplementary Figs. 1 and 6** to include the names of the programs as they appear in this repository.

3. *“I found the section from lines 143 to 182 not particularly clear. After reading it three times I still wasn't sure how many mutants were mapped, what the specific issues were with mutants that couldn't be mapped right away, how this was solved and what the final tally was of mapped vs unmapped mutants.”*

To address this concern, we have extensively edited this section for clarity while also addressing the concerns of reviewer 3 (lines 186-248). In addition, in further response to comment 1 by reviewer 2, we have added clarifying text that also highlights the differences between the approach to combinatorial pooling taken by Knockout Sudoku and that taken by robotic methods. Finally, we have added an additional panel (**I**) to **Fig. 3**.

Response to Reviewer 3

In summary, reviewer 3 comments that they were unable to comment on some of the work due to its incompleteness:

“Main problem to review this work was that documents were incomplete and they did not include any online methods (where everything was hopefully much better explained). Seems that in most places on the manuscript they explain very little and simply referred to online methods. I can still comment something in the methodological scheme (based on the information provided in the actual manuscript): ”

We apologize for the problem, we were under the belief that online methods were included in the original submission review packet from lines 475 to 745 following the suggested submission format. However, in response to specific comments from reviewer 3, we have included additional clarifying material in the main text and supplementary sections.

1. *“Transposon insertions are assumed to be Poisson-distributed (the authors should comment how strong is the assumption that insertions occur independently, which follows directly from Poisson distribution assumption). The authors simply use Poisson probability formula to obtain lower estimate of non-essential gene set.”*

We have added additional material in the main text (lines A-B) (line 101-137) that clarifies how Poisson statistics are used in this study. In summary, the consequences of using the Poisson statistics based model are highlighted here not because it is particularly well justified, but because it is commonly used in most laboratories (line 104) to decide upon transposon mutant collection size. In response to comments by reviewer 1, we have added an additional statements to lines 51-53 that highlight the consequences of this choice of model.

2. *“Monte Carlo simulations are used to obtain disrupted gene count by random draws from the observed set of transposon insertion mutants. They used the most conservative of the two estimates (one given by Poisson formula and one given by MC simulation). Authors should comment why MC was giving larger estimate than Poisson distribution. Explain also what is the rationale of using MC sampling, why it can work here? Explain also Figure 2 much better.”*

We have added additional clarifying text to the Results section (lines 128-137) that explain the greater detail that can be captured in a Monte Carlo simulation of transposon insertion and why it will produce a lower estimate of the fraction of a genome covered by disruptions than a Poisson model for any transposon mutant collection size. In short, a naive poisson estimation neglects the effects of

variable gene sizes, variable transposon insertion site availability as well as the possibility of insertions between coding regions. Additionally, to provide greater clarity to **Fig. 2** we have added three extra panels and clarifying text to the caption.

3. *“Bayesian inference is mention in the abstract but the actual use of Bayesian methods is limited to the use of simple Bayes formula if I understood right (based on minimal explanation provided in the ms), not any complicated MCMC calculations or fancy models. Therefore, “Bayesian inference” in the abstract may be good to replace with “Bayes formula”. Please provide also better explanation of that in the actual manuscript. Currently I was not able to obtain idea of the whole story based on the current text.”*

This point is well taken, and we have changed the description of our algorithm to “probabilistic” throughout the manuscript and supplementary material.

4. *“The part before application of Bayes formula, some numbers to be substituted to formula were given by fit of Voigt function. To me this looked like approximating shape of empirical distribution of something with this function. That parts was very interesting but again in the manuscript there was completely insufficient explanation of the procedure (hopefully online methods did the better job). Please add some explanation also to the manuscript part.”*

Indeed, these numbers were taken from the Voigt approximation of the empirical distribution. To clarify we have added additional explanatory text to the Results section (lines 216-234)

Supplementary Notes 1 to 3 and **Supplementary Figs. 4 and 5**.

Reviewer #2 (Remarks to the Author)

As said before a resource and method like this is useful and valuable and I believe all comments have been sufficiently addressed.

I have three minor points:

1. P3L108-110. Where does the information come from that ~900 genes are essential, ~3500 non-essential and that some regions cannot accept transposons? If it is reference 13, please insert this reference in the beginning of the paragraph and it would probably be useful to explicitly mention what the data is based on.
2. P9L377. Duplication of 'the most'.
3. P9L382-383. I think it's either 'they are' or 'they can be'.

Response to Reviewer 1

Reviewer 1 comments:

“I have found the response to my specific questions and concerns by the authors to be thorough and satisfactory.”

Response to Reviewer 2

Reviewer 2 comments:

“As said before a resource and method like this is useful and valuable and I believe all comments have been sufficiently addressed.”

Reviewer 2 notes three minor points in the manuscript that require attention;

1. *“P3L108-110. Where does the information come from that ~900 genes are essential, ~3500 non-essential and that some regions cannot accept transposons? If it is reference 13, please insert this reference in the beginning of the paragraph and it would probably be useful to explicitly mention what the data is based on.”*

We have adjusted lines 102 to 112 in the revised manuscript to improve the clarity of this paragraph this comment. Deutschbauer *et al.* (reference 13 in the current and previous versions of the manuscript) used transposon sequencing to estimate the essentiality of the 3,659 of 4,587 genes in the

S. oneidensis genome. We have moved a reference to this article to the beginning of this paragraph. Of the 4,587 genes in the *S. oneidensis* genome, 403 were marked as definitively essential, while 3,256 were marked as definitely non-essential. Meanwhile, the essentiality of 928 was listed as unknown. We have further elaborated on this in the Methods section and have included a reference in this revised paragraph.

The statement that there will be regions of the genome that will not accept transposons (due to a lack or scarcity of the AT/TA transposon insertion site) is, at this point in the manuscript, technically a prediction. It is conceivable that there are some genomes where each and every gene has at least one AT/TA dinucleotide (other than the start and stop codons), but on balance of probability we think this is highly unlikely. However, for clarity we have amended the text to indicate that this is actually a prediction.

2. *“P9L377. Duplication of 'the most'”*

We have corrected this duplication error in line 336 of the revised manuscript.

3. *“P9L382-383. I think it's either 'they are' or 'they can be'”*

We have corrected this grammatical error in line 341 of the revised manuscript.

Response to Reviewer 3

Reviewer 3 comments:

*“The authors have satisfactorily answered to all my criticism.
I do not have any further comments.”*